# LIVE-PAINT allows super-resolution microscopy inside living cells using reversible peptide-protein interactions

Curran Oi [1,2], Zoe Gidden [3], Louise Holyoake[3], Owen Kantelberg[4], Simon Mochrie[2,5], Mathew H. Horrocks [4✉] & Lynne Regan [6✉]

We present LIVE-PAINT, a new approach to super-resolution fluorescent imaging inside live cells. In LIVE-PAINT only a short peptide sequence is fused to the protein being studied, unlike conventional super-resolution methods, which rely on directly fusing the biomolecule of interest to a large fluorescent protein, organic fluorophore, or oligonucleotide. LIVE-PAINT works by observing the blinking of localized fluorescence as this peptide is reversibly bound by a protein that is fused to a fluorescent protein. We have demonstrated the effectiveness of LIVE-PAINT by imaging a number of different proteins inside live *S. cerevisiae*. Not only is LIVE-PAINT widely applicable, easily implemented, and the modifications minimally perturbing, but we also anticipate it will extend data acquisition times compared to those previously possible with methods that involve direct fusion to a fluorescent protein.

[1] Department of Molecular Biophysics and Biochemistry, Yale University, New Haven, CT 06520, USA. [2] Integrated Graduate Program in Physical and Engineering Biology, Yale University, New Haven, CT 06520, USA. [3] School of Biological Sciences, University of Edinburgh, Edinburgh EH9 3DW, Scotland. [4] School of Chemistry, The University of Edinburgh, Edinburgh EH9 3FJ, Scotland. [5] Department of Physics, Yale University, New Haven, CT 06520, USA. [6] Center for Synthetic and Systems Biology, Institute of Quantitative Biology, Biochemistry and Biotechnology, School of Biological Sciences, University of Edinburgh, Edinburgh EH9 3BF, Scotland. ✉email: mathew.horrocks@ed.ac.uk; lynne.regan@ed.ac.uk

  1

Optical microscopy has traditionally been restricted to a resolution of ~250 nm due to the diffraction limit of light. New methods, collectively grouped under the term super-resolution microscopy, have increased the resolution of fluorescence microscopy by almost two orders of magnitude, allowing systems previously inaccessible to fluorescence microscopy to be studied[1–5]. These methods rely on either limiting the illumination of the sample to regions smaller than the diffraction limit[1], or stochastically and temporally separating the emission of individual fluorophores to allow their positions to be precisely localized. This latter strategy is termed single-molecule localization microscopy (SMLM), and various approaches have been developed to enable the required stochastic emission, including stochastic optical reconstruction microscopy (STORM)[5], photoactivation localization microscopy (PALM)[2], and point accumulation for imaging in nanoscale topography (PAINT)[6].

In the original implementation of PAINT, fluorescent molecules (for example, Nile red) bind transiently and non-specifically to hydrophobic regions of a structure[6], and a super-resolution image is built up as each one is localized. Unlike PALM and STORM methods, which are limited by photobleaching of the dye molecules over time, in PAINT-based methods there is continual replenishment of the fluorescent probes, which allows much longer imaging times, resulting in a higher density of localizations, and the potential for a higher resolution image[7]. In all PAINT methods, the concentration of the interacting fluorescent molecule can be varied and optimized.

DNA-PAINT was developed to enable specific and finely tunable binding of the fluorophore to the structure to be imaged[8]. It relies on covalently attaching a short oligonucleotide sequence to the biomolecule of interest. A super-resolution image is built up as fluorescently labeled DNA oligonucleotides of complementary sequence to the oligonucleotide attached to the biomolecule of interest, transiently hybridize with it and thus are localized. DNA-PAINT is attractive because it is relatively straightforward to vary the strength of strand association by varying the sequence or length of the DNA oligonucleotides. The reversibility of duplex formation allows for replenishment of signal at the position of interest by restricting the illuminated volume of the sample, for example, using total internal reflection fluorescence microscopy (TIRFM) or light-sheet fluorescence microscopy. Thus, only fluorophores in the illuminated portion of the sample are bleached and they will be replaced by unbleached fluorophores when the DNA strands dissociate. By having a large reservoir of fluorophores that can exchange with the bleached ones, many localization events can thus be captured, enabling very high-resolution images to be collected. Localizations with low precision can be discarded, which also contributes to increased resolution.

DNA-PAINT has seen many variations and innovative applications[8–11]. A significant limitation, however, is that DNA-PAINT cannot be used to visualize proteins inside live cells[12]. Although extensions of DNA-PAINT, in which the DNA is fused to a nanobody or another protein-binding module, enable intracellular proteins to be visualized, the cell must be permeabilized to allow them to enter. As a result, work in live cells has been limited to the visualization of cell-surface proteins[12].

Here, we describe a PAINT-based method that has all the advantages of DNA-PAINT, but with the enormous benefit that it can be used for imaging inside live cells. We refer to this approach as LIVE-PAINT (Live cell Imaging using reVersible intEractions PAINT). In LIVE-PAINT, reversible peptide–protein interactions, rather than zipping/unzipping of a DNA oligonucleotide duplex, are responsible for the transient localizations required for SMLM. The protein to be imaged is genetically fused to a short peptide and expressed from the protein's endogenous promoter.

Additionally, integrated at a suitable place in the genome, a peptide-binding protein is genetically fused to a fluorescent protein and expressed from an inducible promoter, allowing its expression level to be controlled and optimized. The small size of the peptide tags fused to the protein of interest is another important strength of the method. It enables post-translational fluorescent labeling of target proteins that do not tolerate a direct fusion to a fluorescent protein. To illustrate this point, we show that LIVE-PAINT can be used to perform in vivo super-resolution imaging of proteins, such as actin and cofilin, which are notoriously refractory to direct fusions[13,14]. Furthermore, we show that LIVE-PAINT can be used to perform diffraction-limited tracking of individual biomolecules for extended periods of time.

## Results

**Peptide-protein pairs can be used to achieve super-resolution.** The essence of LIVE-PAINT is to visualize individual fluorescent molecules transiently attached to a cellular structure of interest. The individual fluorophores are thus identified by temporal, rather than spatial, separation. LIVE-PAINT achieves sparse labeling by using reversible peptide–protein interactions. The protein of interest is directly fused to a peptide and a fluorescent protein is fused to the cognate protein (Fig. 1a). The peptide–protein interactions are chosen so that solution exchange occurs on a timescale shorter than or comparable to the bleaching lifetime, allowing many sequential images to be obtained. In each image, a different peptide-tagged protein of interest is bound to a different protein-FP, allowing individual proteins to be precisely localized (Fig. 1a–d). These localization events are then summed to generate a super-resolution image (Fig. 1e).

As a test case with which to optimize this approach, we visualized the cell division septin protein Cdc12p, a component of the readily-identifiable septum that is formed during *Saccharomyces cerevisiae* budding. We tested LIVE-PAINT using two different peptide–protein interactions with very different dissociation constants and molecular structures: TRAP4-MEEVF (a tetratricopeptide repeat-peptide pair with a dissociation constant ($K_D$) of 300 nM) and SYNZIP17-SYNZIP18 (an antiparallel coiled coil pair with a $K_D$ of 1 nM)[15–18]. In both cases, the peptide (MEEVF or SYNZIP18) is fused to Cdc12p and the cognate protein or peptide (TRAP4 or SYNZIP17, respectively) is fused to the bright green fluorescent protein mNeonGreen (mNG)[19]. Although mNG is known to blink intrinsically[20], we chose to use it in our experiments because it is very bright and therefore can produce very precise localization events. mNG has a brightness of 93[19], while other fluorescent proteins we use in this work, mKO and mOrange, have brightness of 31[21] and 49[22], respectively. Most importantly, we show that mKO and mOrange, which are not known to blink intrinsically, are also compatible with LIVE-PAINT (Supplementary Fig. 1 and Supplementary movies 1 and 2). TRAP-peptide pairs have been shown previously to be less perturbative for cellular imaging than direct fusion to a fluorescent protein[23]. Both TRAP4-MEEVF and SYNZIP17-SYNZIP18, were well-tolerated by the cell, and both can be used for either diffraction limited or super-resolution imaging of the septum in live yeast (Fig. 1e).

As with other super-resolution imaging methods, resolution improves as more localizations are acquired. In our method, we are able to obtain ~20 nm resolution in ~5 s when imaging Cdc12p-SYNZIP18 using SYNZIP17-mNG (Supplementary Fig. 2). We observed no distorted cell morphology or changes in growth rate in liquid media when using the TRAP4-MEEVF and SYNZIP17-SYNZIP18 interaction pairs. In previous work we observed distorted cell morphology for ~5% of yeast expressing a

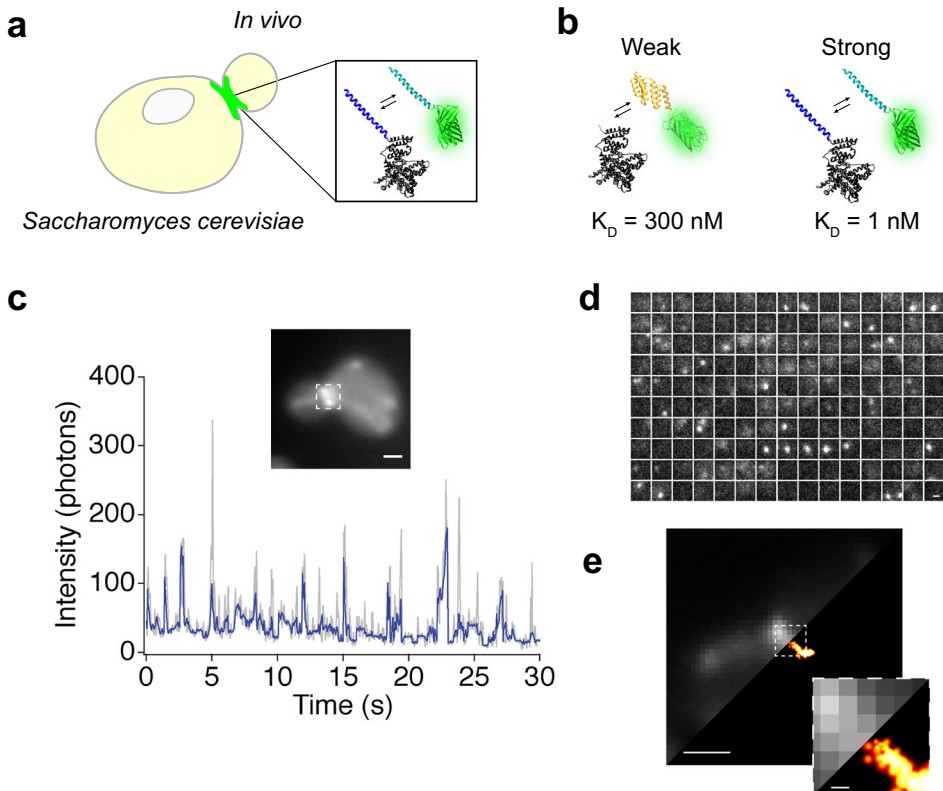

**Fig. 1 LIVE-PAINT achieves sparse labeling using reversible peptide–protein interactions. a** Details of the LIVE-PAINT imaging method, as applied to Cdc12p. A peptide tag (dark blue) is fused to the target protein that will be imaged, Cdc12p (black). A peptide-binding protein (light blue) is fused to a fluorescent protein (bright green). The peptide tag and peptide-binding protein reversibly associate, as indicated by the double arrows. **b** Molecular details of the peptide-protein pairs used: TRAP4 (yellow) binds to the peptide MEEVF (red) and SYNZIP18 (dark blue) binds to SYNZIP17 (light blue) with the dissociation constants shown. Proteins are shown with a ribbon representation of their structures, and are approximately to scale. Ribbon structure diagrams were generated using PDB files for interaction pairs similar to those used in this work: TRAP4-MEEVF is represented using the structure for a tetratricopeptide repeat protein in complex with the MEEVF peptide (PDB ID: 3FWV) and SYNZIP17-SYNZIP18 is represented using the structure for the antiparallel coiled coil Kif21A (PDB ID: 5NFD). **c** Binding and unbinding of the peptide-binding module-fluorescent protein to the peptide tag generate blinking events. Plot of the fluorescence intensity (photons) at a particular location (in a square shown as a dotted box around the septum in the florescence image) in the septum versus time. We interpret peaks in the signal as indicating that mNG is bound to Cdc12p and troughs indicating mNG is dissociated from Cdc12p. **d** Montage of frames from a fluorescence microscopy video collected of the area of the septum boxed in part (**c**). Each frame in the montage is separated by 0.2 s and the bright blinking events correspond to fluorescence peaks in **c**. **e** Diffraction limited (left) and super-resolution (right) images of Cdc12p imaged using Cdc12p-SYNZIP18 and SYNZIP17-mNG. The image was generated from a video with 6000 frames, with an exposure time of 50 ms per frame and a laser power density of 3.1 W/cm$^2$. Number of super-resolution localization events: 448. Scale bars are 1 µm, except for the inset to **e**, which has a 100 nm scale bar.

direct fusion of Cdc12p to a fluorescent protein[23]. We also provide evidence that LIVE-PAINT can be performed with additional peptide–protein interaction pairs (Supplementary Fig. 3), that the localizations events observed are specific to the protein being labeled (Supplementary Fig. 4), and that two orthogonal interaction pairs can be used with two fluorescent proteins to tag two different proteins specifically and concurrently (Supplementary Fig. 5).

**Signal to background dictated by amount of labeling protein.**
In LIVE-PAINT, the peptide-binding proteins fused to mNG (TRAP4-mNG and SYNZIP17-mNG), are expressed from an inducible promoter, so that expression levels can be optimized[24]. See Supplementary Fig. 6 for fluorescence induction profile.

By varying the expression level of either TRAP4-mNG or SYNZIP17-mNG, for the TRAP4-MEEVF and SYNZIP17-SYNZIP18 interaction pairs respectively, we can determine which conditions generate the highest percentage of localizations at the septum relative to non-specific localizations (Fig. 2). For very low

expression levels, for example for 0% galactose with 'leaky' expression, not enough mNG is expressed and not enough localization events are achieved to generate a super-resolution image. Conversely, for example for 0.1% galactose, expression levels are too high and very few individual localization events can be visualized, because the density of mNG is too high to achieve sparse labeling. At intermediate expression levels, for example with 0.005% or 0.02% galactose, there are sufficient fluorescent proteins that enough localization events can be recorded to resolve a super-resolution image, but the fluorescent protein expression level is not so high that single localization events cannot be recorded.

We performed cluster analysis using the DBSCAN function (see Methods section) to quantify the number of localization events in the septum versus in the rest of the cell. We were thus able to identify the conditions that produced the most specific super-resolution images. In an analogous fashion to DNA-PAINT, the fluorescent protein mNG does not give rise to a localization event until it binds and is immobilized. Some non-specific localization or blinking events are recorded, these are

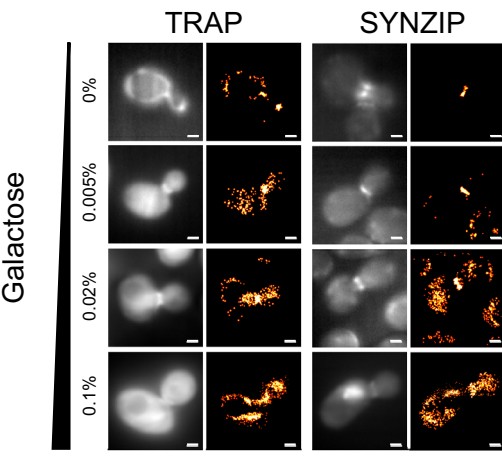

**Fig. 2 Varying either the fluorescent protein expression level or the peptide–protein interaction pairs changes the number of localization events, which are specifically localized to the yeast bud neck during cell division.** Pairs of diffraction-limited and super-resolution images are shown for Cdc12p-MEEVF + TRAP4-mNG (left) and Cdc12p-SYNZIP18 + SYNZIP17-mNG (right), at different concentrations of galactose (as indicated on the left). All images were generated from 1000 frame videos, with each frame having an exposure time of 50 ms and a laser power density of 3.1 W/cm². Percent of localizations in septum, at different concentrations of galactose, for Cdc12p-MEEVF + TRAP4-mNG: 0% galactose - 15%; 0.005% galactose - 45%; 0.02% galactose - 38%; and 0.1% galactose - 23%. Percent of localizations in septum, at different concentrations of galactose, for Cdc12p-SYNZIP18 + SYNZIP17-mNG: 0% galactose - 94%; 0.005% galactose - 98%; 0.02% galactose - 43%; and 0.1% galactose -19%. See Supplementary Table 1 for total number of localizations per image. Scale bars are 1 μm.

randomly distributed within the cell and can be removed from further analysis by using DBSCAN. The number of these non-specific localization events increases with galactose concentration, because by increasing galactose we increase the number of free mNG which are not bound to a Cdc12p. For this reason, we choose not to work with very high galactose concentrations for most of our experiments. We observed that the highest percentage of localization events in the septum for the 0.005% galactose condition when imaging both the TRAP4-MEEVF interaction pair (45% of localization events in the septum) and the SYNZIP17-SYNZIP18 interaction pair (98% of localization events in the septum). The septum was identified by cluster analysis (see Methods section).

**Septum width increases as daughter to mother ratio increases.** To demonstrate the potential of LIVE-PAINT, we show an example of how it can be used to study a biological structure in live cells. By analyzing SMLM data for Cdc12p in individual cells, obtained using LIVE-PAINT with the SYNZIP17-SYNZIP18 interaction pair, we are able to describe various features of the yeast budding process. For example, we find that for small daughter cell sizes (daughter: mother diameter ratio <~0.85), the septum width is of the order 200 nm. As the daughter cell gets larger (daughter:mother diameter ratio ~0.85–1.0), the septum is clearly visible as two separate rings, with a septum width of ~400–800 nm. See Supplementary Fig. 7. This example demonstrates that LIVE-PAINT can be used to study a biological structure in live cells on the single cell level.

**Multiple tandem mNG improves localization precision.** In current super-resolution imaging techniques used inside live cells,

such as PALM, the target protein is directly fused to a fluorescent protein. This fusion adds a large modification (25 kDa) to the target protein. Trying to enhance the PALM signal by fusing three fluorescent proteins to the same target protein would increase the size of the overall protein by ~75 kDa. Many proteins are unable to fold and correctly mature to their functional state when fused to a single fluorescent protein, therefore a larger modification to a target protein, on the order of 75 kDa would likely be even more detrimental.

With the LIVE-PAINT method, however, the protein of interest is labeled post-translationally and reversibly. Thus, labeling with multiple tandem fluorescent proteins should be more feasible. We performed LIVE-PAINT on Cdc12p-SYNZIP18 using the SYNZIP17 fused to one or three tandem copies of mNG and compared the super-resolution data obtained for both conditions (Supplementary Figs. 8 and 9). Cdc12p not only tolerates such post-translational labeling with the three tandem mNG, but labeling with this construct results in better localization precision. We note, however, that the larger size of the three tandem mNG construct creates additional distance between the protein of interest which would result in increased uncertainty about its actual position.

**LIVE-PAINT enables longer data acquisition times.** An additional advantageous feature of the LIVE-PAINT method is that it allows bleached fluorescent labels to exchange with unbleached fluorescent labels, in vivo. In the case of STORM and PALM imaging methods, photobleaching of the probe adds a limitation to the number of emitters that can be localized. This photobleaching reduces the resolution of the image because it limits the density of emitters that can be measured. Thus, researchers have to resort to using localization events with lower signal to noise than is optimal. In many cases, control of the emission is difficult to achieve, and much of the fluorescent probe is bleached early in the acquisition when individual emitters cannot be discerned due to their density being too high, further limiting the density of localizations measured. Here we demonstrate the ability to image for longer periods of time with LIVE-PAINT, using the SYNZIP labeling pair.

When imaging using a conventional direct fusion of Cdc12p to mNG, we observe that after we deliberately photobleach by irradiating with high laser power for 2 min, very few localization events are subsequently observed. In contrast, when using SYNZIP17-SYNZIP18 to localize mNG to Cdc12p, after we deliberately photobleach by irradiating with high laser power for 2 min, we subsequently observe many more new localization events, indicating that the bleached SYNZIP17-mNGs can unbind and be replaced by unbleached SYNZIP17-mNGs from the cytoplasm. This result shows that the LIVE-PAINT imaging strategy allows one to obtain more total localization events during an imaging session, because they allow for longer imaging times (Fig. 3). The individual cells imaged using LIVE-PAINT for the data in Fig. 3 were measured to have a resolution of ~20 nm (see Supplementary Fig. 10 for maximum projection images of the individual cells analyzed in Fig. 3).

**Increasing exchangeable label extends data acquisition times.** The data in Fig. 3 shows that reversible interaction pairs can unbind from the target protein and signal can be replenished by free protein-mNG binding to the target protein.

Building on this result, we compared how long data collection can be continued, when there is a high versus low level of peptide-binding protein-mNG in the cytoplasm. Figure 4 shows the results of such experiments for both the SYNZIP17-SYNZIP18

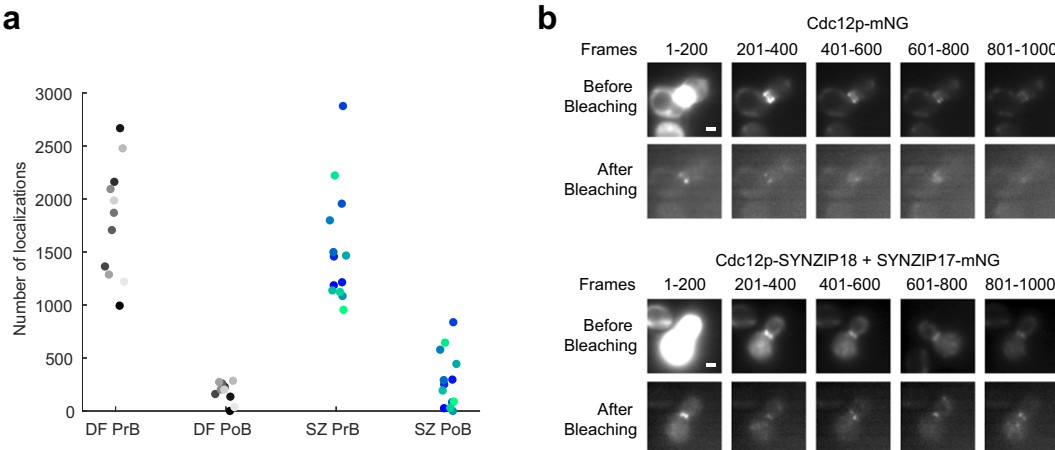

**Fig. 3 LIVE-PAINT shows recovery of signal after bleaching. a** LIVE-PAINT interaction pairs show more recovery in number of localization events than a direct fusion to a fluorescent protein. In this experiment, fluorescence images were collected for 1000 frames (50 s) at standard imaging power (3.1 W/cm$^2$), then the sample was photobleached using high laser power (26.6 W/cm$^2$), and then the sample was again imaged for 1000 frames (50 s) at standard imaging power. Cdc12p-SYNZIP18 + SYNZIP17-mNG (blue/green circles, each representing a single cell) retain many more localization events than Cdc12p-mNG (gray circles, each representing a single cell) after 2 min of photobleaching. Each shade of gray or blue/green represents a single cell, which can be color-matched between pre-photobleaching (PrB) and post-photobleaching (PoB) conditions. DF = Cdc12p-mNG (Direct Fusion); SZ = Cdc12p-SYNZIP18 + SYNZIP17-mNG (SYNZIP pair). **b** Maximum projections for different frame ranges in both "before bleaching" and "after bleaching" videos demonstrate that signal obtained after bleaching continues to localize to the yeast septum. (Top) Maximum projections are shown for 200 frame ranges for a representative cell expressing Cdc12p-mNG. (Bottom) Maximum projections are shown for a representative cell expressing Cdc12p-SYNZIP18 + SYNZIP17-mNG. All "before bleaching" images are normalized to one another and, similarly, all "after bleaching" images are normalized to one another. Scale bar is 1 μm.

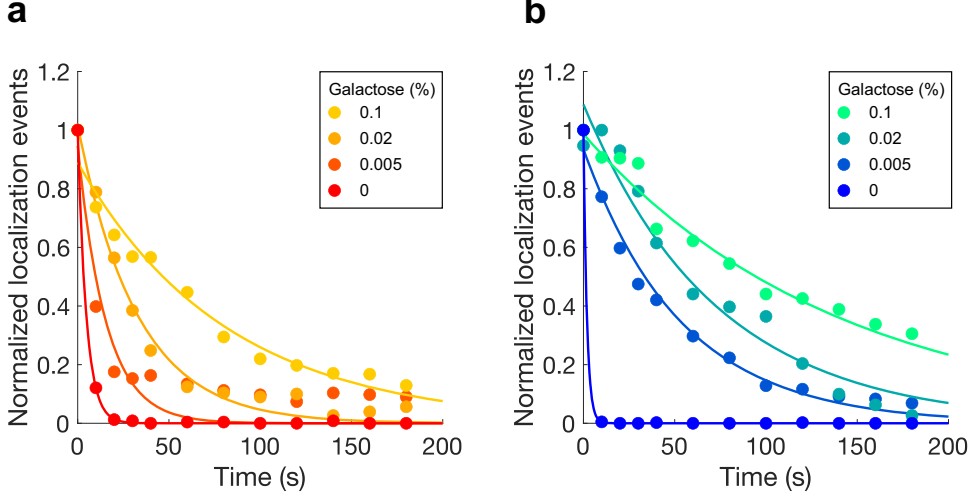

**Fig. 4 Localization rate decays more slowly with increased fluorescent protein expression.** Localization rate as a function of imaging time for **a** Cdc12p-MEEVF + TRAP4-mNG and **b** Cdc12p-SYNZIP18 + SYNZIP17-mNG, each at four different concentrations of galactose. Data for the MEEVF-TRAP4 interaction pair is for 0% galactose (red), 0.005% galactose (dark orange), 0.02% galactose (light orange), and 0.1% galactose (yellow). Data for the SYNZIP18-SYNZIP17 interaction pair is for 0% galactose (bright blue), 0.005% galactose (blue), 0.02% galactose (teal), and 0.1% galactose (mint). All images were generated from 4000 frame videos, with each frame having an exposure time of 50 ms and a laser power density of 3.1 W/cm$^2$. The data for each concentration of galactose were fit to a single exponential (shown as a solid line with matching color). For the MEEVF-TRAP4 interaction pair (**a**), the exponential time constant (τ) for the different concentrations of galactose is 0%: 4.7 s; 0.005%: 15 s; 0.02%: 32 s; and 0.1%: 81 s. For the SYNZIP18-SYNZIP17 interaction pair (**b**), the exponential time constant (τ) for the different concentrations of galactose is: 0%: 1.9 s; 0.005%: 54 s; 0.02%: 73 s; and 0.1%: 139 s.

and TRAP4-MEEVF interaction pairs. For 0% galactose, where the expression level of peptide-binding-module-fluorescent protein is low, almost all the binding-module-fluorescent protein will be initially bound to Cdc12p, thus all fluorescent proteins will be illuminated and bleached rapidly, because there is not a cytoplasmic pool of peptide-binding protein-fluorescent protein for them to exchange with. By contrast, for 0.1% galactose, where

the expression level of the peptide-binding protein-fluorescent protein is high, there is a sizeable cytoplasmic pool available to exchange with molecules bound to peptide-Cdc12p, but which have been bleached. In Fig. 4b, for example, we observe that when imaging Cdc12p-SYNZIP18 + SYNZIP17-mNG using 0.1% galactose, even after 200 s of imaging, localizations are still being recorded at ~30–40% of the initial rate.

**Difficult to tag proteins can be labeled using LIVE-PAINT.**
Actin (Act1p), an important cytoskeletal protein, is notoriously difficult to tag and image. A number of different methods have been developed to circumvent this problem, but they are not without issues, including changing the stability, dynamics, and lifetime of Act1p structures[13,25,26]. Direct fusion of Act1p to the photoconvertible fluorescent protein mEos, expressed alongside unmodified Act1p, has been used to image Act1p using PALM[27,28]. The mEos protein is a rather large addition to Act1p, and undoubtedly results in some perturbation of function (as evidenced by cells expressing only Act1p-mEos, in the absence of any unmodified Act1p, being unviable). LifeAct is a peptide that binds to the polymerized form of Act1p, and not the unpolymerized form. The perturbation to the equilibrium distribution of Act1p forms that LifeAct causes has been noted[25]. Nevertheless, the binding and unbinding of LifeAct has been used to image Act1p filaments in live cells using a PAINT-like approach[29]. We note and reference this result, however polymerized Act1p is the only protein that can be imaged using LifeAct, our method can be applied to any protein, including Act1p, and we present its application to Act1p to provide another possible tool for actin researchers.

Wild-type Act1p was chromosomally expressed from its endogenous promoter. We expressed SYNZIP18-Act1p from a low copy number plasmid, using a copper-inducible promoter. SYNZIP17-mNG was expressed, as previously, from the galactose inducible promoter, chromosomally integrated at the *GAL2* locus (Fig. 5).

Using LIVE-PAINT, we were able to readily visualize actin patches, which assemble at the cell membrane, at sites of endocytosis[30] (Fig. 5a). Because actin structures are quite dynamic, we investigated how quickly we could obtain super-resolution images (compared to the acquisition time of 200 s for the data shown in Fig. 5c). Actin rings, or actin cables that span the cell, are likely not observed because we are imaging in TIRF, which illuminates only ~200 nm into the cell (a typical yeast cell is 1–3 μm thick). Alternatively, or additionally, it could be that the stringent structural requirements for actin in these assemblies means that even actin with very small ~2 kDa tags may be excluded from ring and cable structures[31].

**LIVE-PAINT enables long tracking times in vivo.** In the data presented so far, we have used LIVE-PAINT to generate super-resolution images of proteins which do not move significantly

during the period of data acquisition. In some cases, the protein-of-interest may move on the timescale of imaging, and whilst an increase in the imaging frame rate could resolve this to some extent, it may not always be possible if the proteins move too quickly. The extended imaging lifetime enabled by LIVE-PAINT, however, offers the opportunity to detect and track the motion of diffusing molecules within live cells. Cofilin (Cof1p) is an important protein that binds to actin filaments promoting severing[14]. It has so far, however, proven difficult to image due to its function being affected by either N- or C-terminal direct fusion of a fluorescent protein[14]. We therefore C-terminally tagged Cof1p with SYNZIP18, and tracked it using the LIVE-PAINT strategy (diffraction-limited, not super-resolution). We were able to observe the diffusion of Cof1p during the 100 s of imaging (Fig. 6 and Supplementary movie 3). We observed a wide range of behaviors (Fig. 6).

The success of the LIVE-PAINT tagging approach in these examples demonstrates the value of the method for visualizing proteins that are refractory to direct fusion to a fluorescent protein[14], and also its potential to be developed to track moving proteins.

## Discussion

We have developed an imaging strategy, LIVE-PAINT, which enables a new approach to super-resolution imaging inside live cells. We have demonstrated the effectiveness of LIVE-PAINT, which makes use of reversible protein–peptide interactions to obtain SMLM in live *S. cerevisiae*. The data we obtained for Cdc12p enabled us, for example, to quantitatively track the width of the septum at the bud neck of budding yeast as a function of daughter:mother cell diameter ratio and showed that septum width does not change significantly until the daughter diameter reaches ~0.85 of the mother cell diameter, at which point the septum divides into two distinct rings.

LIVE-PAINT has a number of advantages over existing super-resolution imaging methods. The main advantage over DNA-PAINT is that LIVE-PAINT works inside living cells; all the components that we describe are chosen to function in that *milieu*. Also, LIVE-PAINT is extendible to concurrent tagging of multiple proteins. Recently, interacting charged coiled coil pairs have been used to label proteins to perform PALM imaging in live mammalian cells[32]. This work provides a valuable independent validation of our approach.

LIVE-PAINT requires neither photo-conversion of fluorophores (as PALM does) nor selective deactivation of fluorophores as stimulated emission depletion (STED) does. Not only do these methods require special instrumentation, but the high laser power that is typically required can often cause cell damage during live cell imaging experiments, in addition to bleaching of fluorophores[33]. LIVE-PAINT is performed inside living cells, typically in minimal growth medium, with no potentially toxic additions, such as oxygen scavengers, required. LIVE-PAINT requires only that the protein of interest is directly fused to a small peptide tag, a strategy with a number of advantages. Labeling is post-translational, and therefore the method is suitable for labeling proteins for which direct fusion to a larger fluorescent protein abrogates function[23]. Other approaches to performing PAINT in live cells, such as protein-PAINT, require the addition of organic dyes, cannot be used to image multiple targets simultaneously, and require a larger fusion to the target protein[34].

The intensity of the signal from each localization event can be increased, by using a tandem array of fluorescent proteins attached to the peptide-binding protein. It is also straightforward to change the identity of the fluorescent protein, without needing

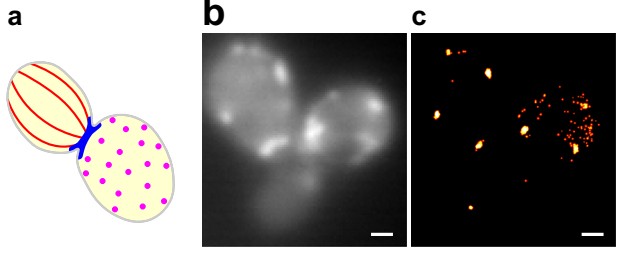

**Fig. 5 Actin patches can be imaged using LIVE-PAINT in live yeast.**
**a** Cartoon showing the three distinctive actin structures that have been observed in fixed and immunostained *S cerevisiae*: actin cables (red), actin rings (blue), and actin patches (magenta). **b** Diffraction limited image of SYNZIP18-Act1p + SYNZIP17-mNG. **c** LIVE-PAINT super-resolution image constructed from 200 s video imaging SYNZIP18-Act1p + SYNZIP17-mNG (50 ms exposure per frame and a laser power density of 3.1 W/cm²). Number of localization events obtained: 778. Only localization events with precision <30 nm were used to construct the super-resolution image. Scale bars are 1 μm.

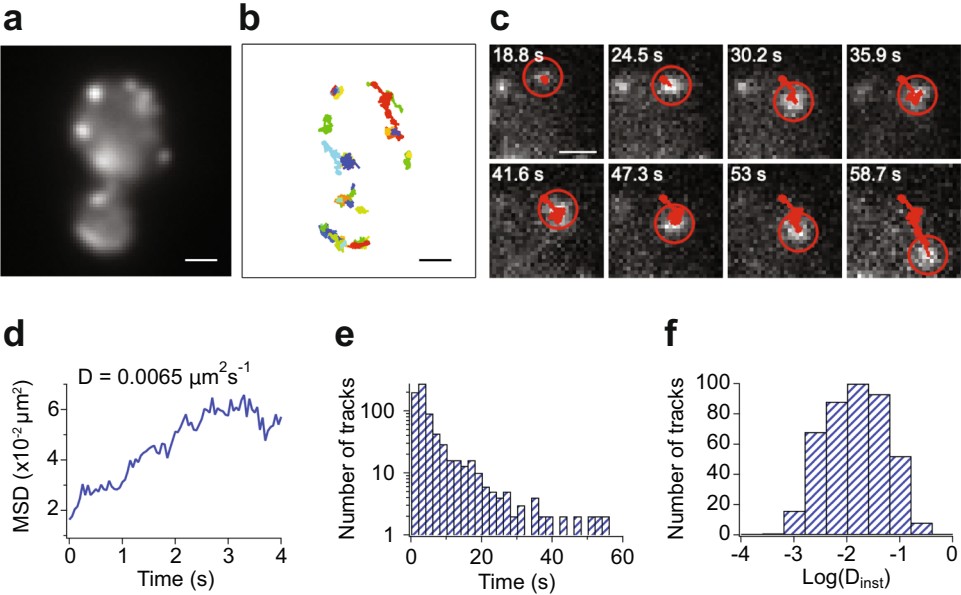

**Fig. 6 Clusters of Cof1p can be tracked using LIVE-PAINT. a** Diffraction-limited image of example live yeast in which Cof1p was tagged and tracked. Larger field of view shown in Fig. S11. **b** Individual tracks of Cof1p from the cells shown in **a**. The yeast cells were imaged for 100 s (50 ms per frame) and a laser power density of 3.1 W/cm$^2$, and each colored track corresponds to an individual diffusing cluster (see Supplementary Movie 3). **c** Example montage of one of the diffusing Cof1p clusters (see Supplementary Movie 4). **d** The mean squared displacement of the tracked Cof1p cluster shown in **c**. **e** Histogram of track lengths from all clusters detected in the cells in Fig. S11. The tracks last for 6.59 ± 9.83 s (mean ± S.D., $n = 426$). **f** Histogram of the diffusion coefficients of the tracked Cof1p clusters. $D = 0.031 ± 0.043 \, \mu m^2 s^{-1}$ (mean ± S.D., $n = 426$). Scale bars are 1 μm.

to change the peptide fusion to the protein to be imaged. Because LIVE-PAINT does not rely on the use of photoactivatable proteins, any fluorescent protein can be used. This flexibility in choice of fluorescent protein means that the method could be extended to concurrent super-resolution imaging of multiple targets. One of the limitations of existing live cell super-resolution methods is the reliance on fluorescent proteins that are all very spectrally similar to one another, which prevents accurate imaging of multiple target proteins concurrently. Even though recent methods have been extended to image two targets in live cells, they require harsh oxygen scavengers and are limited to only two colors by the lack of additional orthogonal chemistries for attachment of dyes to protein tags such as SNAP-tag[35]. LIVE-PAINT does not require oxygen scavengers and is limited only by the number of orthogonal peptide–protein interaction pairs and the number of available spectrally distinct fluorescent proteins, both of which are abundant.

Importantly, the protein of interest is expressed from its endogenous promoter and the conditions for detection of fluorescence localizations are optimized by adjusting the intracellular concentration of the peptide-binding protein-fluorescent protein that is used for labeling. This means that for a very abundant protein of interest, for example, in LIVE-PAINT the number of fluorescent proteins can be reduced by reducing the expression level of the peptide-binding protein-fluorescent protein, instead of having to photobleach some of the fluorescent proteins to reduce the number so that individual fluorescent proteins can be localized. Similarly, for a low abundance protein of interest directly fused to a fluorescent protein, photobleaching is especially problematic, because the starting number of molecules is very low. In LIVE-PAINT, more localization events can be observed by imaging for longer, during which time any bleached peptide-binding protein-fluorescent proteins can be refreshed by exchange with an unbleached pool.

Finally, LIVE-PAINT, especially in a TIRFM (or light-sheet fluorescence microscopy) format, enables data to be acquired for

much longer than other current methods (such as PALM) that can be used inside live cells. In such methods, the fluorescent protein is directly fused to the protein of interest, so once a fluorophore is bleached, it is not replaced and from then onwards is dark. By contrast, in LIVE-PAINT, the non-covalently bound fluorescent protein can be exchanged after bleaching, with a non-bleached fluorescent protein from the cytoplasm. Acquiring data for longer results in more localizations being detected and consequently higher resolution images being obtained. Unbound fluorescent proteins in LIVE-PAINT result in background fluorescence, but this effect can be mitigated by reducing the illumination volume in the cell, as we have done using TIRFM in this work, or by using other strategies, such as light-sheet fluorescence microscopy.

We have demonstrated the power of LIVE-PAINT in *S. cerevisiae* by using it to image Cdc12p and hence to study septum formation. Furthermore, we have used it to image Act1p and Cof1p, two important proteins that are intractable to direct fusion. Finally, we showed that this approach is fundamentally compatible with tracking the movement of individual proteins inside live cells.

We expect that it will be straightforward to extend LIVE-PAINT to other organisms and cell types. In our work we found that two of the three peptide-pairs that we tested were suitable for LIVE-PAINT. Many more potentially compatible interaction pairs exist, and may be better suited for particular applications. In future work, we will investigate how the optimal labeling requirements differ for different cellular proteins and how best to label and image multiple proteins simultaneously.

## Methods
**Molecular biology**. All cloning was performed in *Escherichia coli* strain TOP10. Peptide tags were cloned into pFA6a-KANMX6 by amplifying the plasmid backbone and inserting gBlocks (Integrated DNA Technologies) using NEBuilder® HiFi DNA Assembly Master Mix (New England Biolabs). Except where otherwise noted, the protein sequence used to link different protein components was GGSGSGLQ. The two residue linker, GS, was used between the mNG proteins to create the three

mNG array. The 3xmNG construct itself was joined to SYNZIP17 using our standard GGSGSGLQ linker.

Peptide-binding proteins fused to fluorescent proteins were cloned into the pFA6a-HIS3MX6 and tagged Act1p constructs were cloned into the pCu415CUP1 vector (CEN6/ARS4 origin of replication) using the methods referenced above.

The linker used to fuse Act1p to SYNZIP18 or MEEVF was GGSGSG.

Primer sequences used in this study are listed in Supplementary Tables 2, 3, and 4.

**Peptide–protein interaction selection**. TRAP4-MEEVF[15], SYNZIP17-SYNZIP18[17], and CC-A$_N$[3.5]-CC-B$_N$[3.5][36] interaction pairs were identified in the literature and tested in vivo in live yeast by fusing one half of the interaction pair to a protein of interest and the other half to a fluorescent protein. The cells were imaged under a microscope as described in the "Microscopy" section. The TRAP4-MEEVF and SYNZIP17-SYNZIP18 interaction pairs showed no morphological abnormalities, no noticeable growth defect in liquid culture or on plates, and showed fluorescence at the expected locus of the tagged protein. For this reason, these two interaction pairs were used in this work. Using CC-A$_N$[3.5]-CC-B$_N$[3.5] interaction pair resulted in unusual cell morphology in the vast majority of cells, with elongated cell shapes, so this pair was not used further. The 101A-101B and 108A-108B interaction pairs[18] were also checked and did not cause any changes to cell morphology or changes to cell growth.

**Yeast strain construction**. Except where otherwise noted, standard methods for genetically modifying yeast and preparing growth media were used[37]. These methods are described below.

The yeast strains and selection markers used in this study are listed in Supplementary Table 5. Yeast strains constructed in this study are all derived from the parent strain BY4741.

C-terminal tags were amplified from pFA6a-kanMX6 yeast integration vectors, along with the KanR marker. The amplification primers also included 45 bp homology arms, which matched the final 45 bp preceding the stop codon in the protein to be tagged and 45 bp downstream of the stop codon.

Transformants were selected by plating first on YPD plates and then replica plating to yeast agar plates including 600 mg/L geneticin (Gibco) and incubating for a further 48 h.

Fluorescent protein fusions were inserted into the yeast genome at the *GAL2* locus by amplifying the desired protein's sequence from a plasmid. The amplification primers also included 45 bp homology arms that match sequences upstream and downstream of the *GAL2* gene, and the *HIS3* gene.

Transformants were selected by plating on synthetic complete agar plates lacking histidine. Strain construction was verified by PCR amplification of the modified locus (using primers from Supplementary Table 2).

First, genomic DNA (gDNA) was isolated from colonies obtained from yeast transformations. This was done by resuspending a single yeast colony in 100 μL of 0.2 M LiAc + 1% SDS. This mixture was incubated at 75 °C in a heat block for 5 min. Afterwards, 300 μL 100% ethanol was added to the mixture and centrifuged at 15,000 × *g* for 3 min. The resulting pellet was washed with 100 μL 70% ethanol, the ethanol was removed, and the pellet allowed to dry for 10 min. The dried pellet was then dissolved in 20 μL TE buffer (10 mM Tris pH 8.0, 1 mM EDTA pH 8.0) and centrifuged at 15,000 × *g* for 15 s. The supernatant containing the gDNA was then transferred to a fresh tube.

This gDNA was then used as a template for a PCR reaction to confirm the presence of the inserted DNA at the genomic locus being checked. The PCR reactions were carried out using Phusion High-Fidelity DNA Polymerase and GC buffer (New England Biolabs) in a ProFlex PCR system (Thermo Fisher Scientific). All PCR reactions to validate the insertion of DNA at genomic loci ran for 30 cycles and used an extension time of 2 min and a volume of 10 μL. The annealing temperatures used for checking each locus varied: *CDC12* (59 °C), *GAL2* (64 °C), and *COF1* (61 °C). PCR products were run on a 1% agarose gel in TAE buffer (40 mM Tris, 20 mM acetic acid, and 1 mM EDTA pH 8.0) for 30 min at 120 V.

**Microscopy**. For imaging experiments, yeast cells were grown overnight in 500 μL of synthetic complete media. Constructs using the *GAL1* promoter were all grown with 1% w/v raffinose plus the concentration of galactose desired for a particular experiment. The concentration of galactose used varied between 0% and 2% w/v.

One colony was picked into a 500 μL overnight culture to ensure that the OD$_{600}$ of the cells was between 0.1 and 0.5 by the time of imaging. Two dilutions of the overnight culture, 1:1 and 1:5, were prepared to ensure that one would fall in this OD$_{600}$ range.

In all, 22 × 22 mm glass coverslips with thickness no. 1 (VWR) were cleaned by a 20 min exposure in a 2.6 L Zepto plasma laboratory unit (Diener Electronic). Frame-Seal slide chambers (9 × 9 mm², Biorad, Hercules, CA) were then secured to a coverslip. The surface was prepared for the attachment of yeast cells by coating the surface with 2 mg/mL concanavalin A (Sigma-Aldrich), which was dissolved in PBS pH 7.4, using ~100 μL per well. After leaving the concanavalin A on the surface of the slide for 30 s, it was removed using a pipette tip and by tilting the slide to ensure all liquid was removed. Then, 150 μL of prepared yeast culture was pipetted onto the slide. The yeast culture was left to sit on the slide for ~5 min. The

cells were then aspirated from the slide, the surface washed with milliQ water three times, and then 150 μL fresh milliQ water was then added to the slide before imaging.

Single-molecule imaging was performed using a custom-built TIRF microscope, which restricts the illumination to within 200 nm of the sample slide. ImageJ was used to collect images and videos. The fluorophores were excited with 488 nm illumination. Collimated laser light at a wavelength of 488 nm (Cobolt MLD 488-200 Diode Laser System, Cobalt, Sweden) was aligned and directed parallel to the optical axis at the edge of a 1.49 NA TIRF objective (CFI Apochromat TIRF 60XC Oil, Nikon, Japan), mounted on an inverted Nikon TI2 microscope (Nikon, Japan). The microscope was fitted with a perfect focus system to autocorrect the z-stage drift during imaging. Fluorescence collected by the same objective was separated from the returning TIR beam by a dichroic mirror (Di01-R405/488/561/635 (Semrock, Rochester, NY, USA)), and was passed through appropriate filters (BLP01-488R, FF01-520/44 (Semrock, NY, USA)). The fluorescence was then passed through a 2.5× beam expander and recorded on an EMCCD camera (Delta Evolve 512, Photometrics, Tucson, AZ, USA) operating in frame transfer mode (EMGain = 11.5 e⁻/ADU and 250 ADU/photon). Each pixel was 103 nm in length. Images were recorded with an exposure time of 50 ms with a laser power density of 3.1 W/cm². The lasers were first attenuated with neutral density filters to reduce the excitation power. The power at the back aperture of the objective lens was measured, and the excitation area determined using tetraspeck beads immobilized on a glass coverslip. The microscope was automated using the open source microscopy platform Micromanager.

For photobleach-and-recovery experiments we first imaged the samples at very high laser power density (26.6 W/cm²). After 1000 frames (50 s) of imaging, this power density was dropped to 3.1 W/cm². The sample was then imaged for another 1000 frames (50 s).

For experiments in which mKO and mOrange were used for imaging, a 561 nm laser (Cobolt DPL Series 561-100 DPSS Laser System, Cobalt, Sweden) was used just as the 488 nm laser was used for mNG imaging experiments. The laser power used was 50 W/cm² when imaging mKO or mOrange.

**Microscope settings/imaging parameters**. Images were analyzed using Fiji (Java 8 2017 release) and single localizations were processed using the Peak Fit function of the Fiji GDSC SMLM plugin, using a signal strength threshold of 30, a minimum photon threshold of 100, and a precision threshold of 20 nm. The precision threshold was sometimes changed to 30 nm, 40 nm, or 1000 nm, in order to obtain the distribution of precision values for all obtained localization events. Supplementary Figure 12 shows a matrix of precision and minimum photons per localization thresholds applied to one stack of images, which helped select the cutoffs used.

**Image resolution calculation**. Image resolution was calculated by first performing cluster analysis using DBSCAN[38] in Python 2.7 to identify localizations in the yeast bud neck. Then, resolution was measured using the equation $R_{eff} = \sqrt{(\bar{r}_{nn})^2 + (\bar{\sigma})^2}$, where $R_{eff}$ is the effective image resolution, $\bar{r}_{nn}$ is the mean nearest neighbor distance between localizations in the septum, and $\bar{\sigma}$ is the average localization precision[39].

**Cluster analysis for identifying yeast septum**. For the images shown in Fig. 2, septum localizations were identified from total cellular localization events using DBSCAN[38] in Python 2.7 and the percent of total cellular localizations in the septum was determined. In order to prevent misidentification of septa in background localizations, DBSCAN was applied to localizations within a 1-μm radius of the center of the cell. DBSCAN parameters were maintained for images of cells the same galactose concentration: 0% galactose – $\varepsilon = 2$, $N = 25$; 0.005% galactose – $\varepsilon = 2$, $N = 50$; 0.02% galactose – $\varepsilon = 1.75$, $N = 50$; and 0.1% galactose – $\varepsilon = 2.8$, $N = 75$.

**Quantifying septum width**. Budding yeast with septa were identified from z-projections and following thresholding, ImageJ's Analyze Particles tool was used to determine: the maximum Feret's diameter of the cell, the starting coordinates of the Feret's diameter, the angle between the Feret's diameter and the x-axis, and the coordinates of the cellular center of mass. The end coordinates of the Feret's diameter were calculated from the Feret's diameter data. In the same cells, septum localizations were identified from total cellular localization events as described in above cluster analysis within a radius of the Feret's diameter/5 from the cell's center of mass and using parameters of $\varepsilon = 2$, $N = 100$.

The distance between the center of the septum points and the coordinates of both the start and end of the Feret's diameter was determined and the larger of the two was taken to be the mother cell diameter and the smaller, the daughter cell diameter.

To find the septum width, the mean absolute perpendicular distance between all the septum localizations and the line bisecting the angle between the center of the septum, and the mother and daughter diameters was doubled.

**Plate reader measurements**. Plate reader measurements were carried out on a POLARstar Omega microplate reader (BMG LABTECH). To observe the galactose-dependent induction of mNG under the *GAL1* promoter in a *gal2Δ* background, budding yeast cells were grown overnight in 500 µL of synthetic complete media plus 1% w/v raffinose and galactose concentrations ranging from 0 to 0.1% w/v.

The next morning, 200 µL of this culture was added to individual wells in a 96-well clear bottom plate (Greiner bio-one, item 655096). Cellular fluorescence was excited using the 485 nm excitation filter and measured using the 520 nm emission filter.

The optical density of the cells was measured using the absorbance setting at 600 nm. The fluorescence readings were then normalized to the number of cells by dividing the measured cellular fluorescence by the optical density.

**Single-molecule tracking analysis**. The LIVE-PAINT images were recorded at a frame rate of 50 ms for 2000 frames. The images were first analyzed using Trackpy[40]. Individual puncta corresponding to Cof1p clusters were selected by applying a mask size of 7 and a minimum mass of 2000. The puncta were linked into tracks by applying a maximum displacement of three pixels/frame, and a memory of three frames (i.e. if the puncta were absent in more than 3 frames, then they were no longer linked to those in previous or subsequent frames). Tracks shorter than 20 frames were discarded, and the mean squared displacement (MSD) plot for the remaining tracks were calculated.

Igor Pro (Wavemetrics) was used to calculate the initial diffusion coefficient for each track by fitting the first 250 ms of the MSD to a straight line and determining the gradient. The log of the diffusion coefficients determined from fits with an $r^2 > 0.5$ were then used to populate the diffusion coefficient histogram. A custom code was also used to generate the tracking movies, and the track figure (Fig. 6B).

**Statistics and reproducibility**. A one-tailed *t*-test was performed to compare whether the mean precision values differed between 1xmNG and 3xmNG imaging experiments. First, the mean precision value of all localization events was calculated. Using two replicates (super-resolution images from different fields of view) for each of 1xmNG and 3xmNG experiments, a *t*-test was then performed using the *t*-test function in MATLAB (2019b release).

**Reporting summary**. Further information on research design is available in the Nature Research Reporting Summary linked to this article.

## Data availability
Data files used to generate all charts and graphs, as well as uncompressed supplementary videos, have been deposited to Edinburgh DataShare at https://doi.org/10.7488/ds/2859[41] and https://doi.org/10.7488/ds/2801[42]. A step-by-step protocol is available at Protocol Exchange: https://doi.org/10.21203/rs.3.pex-1043/v1[43]. All relevant data are available from the corresponding author upon reasonable request.

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

## Acknowledgements

We thank Diana Tokarska and Fatima Rafiq for assistance with cloning. We thank Adele Marston and Vasso Makrantoni for the yeast strain BY4741. We thank Chris Wood, Ella Thornton, and Rossana Boni for reading the manuscript and offering helpful suggestions. We acknowledge support from NIH R01 GM118528; The Yale Integrated Graduate Program in Physical and Engineering Biology; the School of Biological Sciences at the University of Edinburgh; BBSRC EASTBIO Doctoral Training Partnership; the School of Chemistry at the University of Edinburgh; The Euan MacDonald Centre; Dr. Jim Love and UCB Pharma for providing funding for the microscope; and the UK Dementia Research Institute.

## Author contributions

C.O., M.H.H., S.M., and L.R. designed experiments and wrote the manuscript. C.O., Z.G., L.H., O.K., and M.H.H. acquired data. C.O., Z.G., M.H.H., and L.R. analyzed data.

## Competing interests

The authors declare no competing interests.
