## [Peer Review File · Communications Biology]

Reviewers' comments:

Reviewer #1 (Remarks to the Author):

Manuscript by Oi et al. reports on bipartite fluorescent protein labeling systems for live-cell imaging in yeast. The system relies on transient interaction of peptides, one attached to the protein of interest, the other one – to a fluorescent protein. The Authors claim the utility of the approach for super-resolution imaging. The Manuscript presents results on PAINT-like super-resolution in living yeasts with two interacting pairs (TRAP4-MEEVP and SYNZIP17-SYNZIP18) and one fluorescent protein (mNeonGreen).

The Referee finds the results promising. The current version of the Manuscript lacks a recent scientific context and overstates the novelty of the reported results. The scientific presentation can be improved. The Referee is mainly concerned with the choice of the self-blinking fluorescent protein as a reporter and with the high affinity of SYNZIP17-SYNZIP18 pair.

The Referee would like to share the following comments on the Manuscript.

Major

[1.1] novelty and scientific context

The authors have overlooked a recent demonstration of PAINT-like super-resolution and live-cell imaging with reversible interactions of small peptides (see Perfilov 2020, DOI: 10.1007/s00018-019-03426-5). Since the Manuscript under consideration is not the first one to show the applicability of transient peptides for live-cell PAINT-like super-resolution of intracellular proteins, the text should be amended accordingly.

[1.2] To put things into perspective, both the current Manuscript and the one by Perfilov are, in fact, live-cell adaptations of IRIS (image reconstruction by integrating exchangeable single-molecule localization, DOI: 10.1038/nmeth.3466). Other examples of small peptide-based tags for live-cell super-resolution imaging exists for nM affinity range (close to the one of SYNZIP17-SYNZIP18), e.g., DOI: 10.1038/ncomms10372. The super-resolution with the transient binding of diffusing probes is an ancient concept; see review by Molle et al. DOI: 10.1016/j.copbio.2015.12.009 The Referee oppose the introduction of yet another catchy abbreviation to the field (LIVE-PAINT). Besides the lack of novelty (see point 1.1), it does not convey the idea that the system is limited to protein labeling.

[2] Support of some claims by the data may not be sufficient

[2.1] choice of mNeonGreen as a reporter fluorescent protein raises concerns due to its intrinsic blinking.

The authors insist that the observed blinking is a manifestation of transient binding. However, mNeonGreen is robustly blinking all by itself. See original paper by Shaner (DOI: 10.1038/nmeth.2413) and a more recent one (DOI: 10.1016/j.bbrc.2019.11.163).

The Referee doubts that the PAINT-like localizations analyzed throughout the text reflect solely binding events and not blinking dynamics of mNeonGreen.

Please discuss the impact of mNeonGreen blinking on the conclusions of the study.

[2.2] Claim of wide applicability and multi-color labeling

Is the method at all usable with non-blinking or non-photoactivatable FPs for super-resolution? What are other fluorescent proteins suitable?

Please support the claims:

LL421 "...is limited only by the number of orthogonal peptide-protein interaction pairs and the number of available spectrally distinct FPs, both of which are abundant."

LL412-413 "Any FP can be used."

– The Referee believes that these claims are not supported by the data in the current version of the Manuscript.

[2.3] The choice of tight (nM-scale binding) peptide pair is not explained and may conflict with the notion of rapid exchange.

The authors start with:

LL122-124 "The peptide-protein interactions are chosen so that solution exchange occurs on a timescale shorter than or comparable to the bleaching lifetime."

But:

LL134 The peptide pair appears to bind quite tightly ($\sim 1\text{nM}$), which is not in line with the idea of PAINT-like exchange. Please elaborate on the choice of peptide pair.

[2.4] Evidence of exchange is insufficient

The experimental evidence does not convince the Referee of the exchange in the system. Moreover, the Referee challenges the idea that $\sim 1\text{nM}$ K_d allows for such exchange at the imaging timescale. Specifically, the photostability of the mNeonGreen-labeled Cdc12 (conventional fuse) should be assessed in direct comparison (Figure 4). Otherwise, the difference between the curves can be attributed to sampling. See also point [3.6]

Concerning the normalized y-axis of the plot, were the initial localization counts similar? The Referee is also a bit surprised by seeing both λ and τ for the decay.

Minor:

[3.1] Scientific presentation.

The abstract should be rephrased in neutral language. The Referee agreed to review this Manuscript after examining the abstract and was disappointed that none of the points in the last sentence was experimentally supported in the current version of the Manuscript.

Consider:

"widely applicable, easily implemented, and the modifications minimally perturbing, but it also allows extended data acquisition times compared to previously possible with methods that involve direct fusion to a fluorescent protein."

Either provide support for

"Widely applicable."

"Minimally perturbing."

"...data acquisition times compared to... fusion to a fluorescent protein."

Or

Amend the text and abstract to reflect only the results reported in the paper

[3.2] The Referee asks the authors to correct the following inaccuracies in the discussion:

LL391-394 The point of 'main advantage over current small molecule-based PAINT' is not clear. To the best of the Referee's knowledge, protein-PAINT (Ref. 25, DOI: 10.1039/C7SC01628J) works within living cells. Please amend the text accordingly.

[3.3] Please discuss the potential limitations and narrow the area of applicability of the method. The obvious point is the impact of the background fluorescence on the SNR.

[3.4] LL140 "Both TRAP4-MEEVF and SYNZIP17-SYNZIP18, were well-tolerated by the cell." It is common to support these types of statements ("well-tolerated") quantitatively. Consider including cell viability, imaging, or other data in support of this claim.

[3.5] The Referee asks the authors to consider and discuss the possibility of self-dimerization of tested coiled-coils, peptides, or their non-specific interaction with cellular proteins. Provide experimental support for the lack of non-specific interactions, if possible, or an appropriate citation.

[3.6] The Referee likes the idea of Figure 3 very much. However, in its current form, the figure is neither convincing (in comparison with classic FRAP experiments), nor quantitative (linear fit for 'ease of visualization' does not provide a quantitative measure of the change in apparent photostability). Moreover, since the DBSCAN filtering of 'specific' vs. 'non-specific' events was applied to the time series, it would be helpful to see maximum projections of the plotted time slots to convince the reader that only specific localizations are counted in such a plot. The Referee thinks that it would be a great Figure, once additional replicas and quantitative analysis are included.

[3.7] The exact number of independent samples should be stated in figure legends. In cases where the representative image or graph is provided (such as Figure 1C), the exact number of experiments with similar results should be indicated.

[3.8] The Referee has failed to find Supplementary Movies 1 and 2 within the submission files

[3.9] Please include exact measurements and calculations behind the power density values ($\sim W/cm^2$) included in the paper. These values are essential for the design of imaging experiments, yet the literature in the field contains numerous examples of order-of-magnitude errors, uncorrected for years. Preferably, include the power measurements at the sample plane without TIRF, calculated or measured irradiation area (it could be measured by bleaching and then shifting the sample a bit). The Referee expects a bit higher values from Cobolt MLD laser.

[4] Data availability and independent analysis. Please provide raw time series behind the graphs and images (Figures 1, 2, 4, 5, 6) on Figshare.

[4.1] The Referee was not able to provide comments on single-molecule analysis without [4]

Reviewer #2 (Remarks to the Author):

The manuscript by Oi et al. describes a new approach for superresolution and single molecule microscopy. The authors use a variation of PAINT (LIVE-PAINT), which is renowned for its high sampling rates. The complementary oligos of DNA-PAINT are exchanged by small interacting peptides. These peptides can be expressed in live cells and are, similar to DNA-PAINT, replenished for a theoretically unlimited number of imaging rounds. The authors use tetratricopeptide repeat affinity proteins (TRAPs), which were used by the lab for live imaging before and synthetic peptides derived from bZIP leucine zippers (Grigoryan et al. 2009).

The authors use yeast for their experiments and perform live imaging of septin, actin and cofilin. They provide evidence that their LIVE-PAINT is suitable for Live-SMLM of proteins with a slow to fast turnover. Although the presented yeast images might not be spectacular at first glance, the method has great potential for a multitude of applications. The modification and adaptation of this method to other cell types (e.g. mammalian cells) will be highly relevant for many researchers.

With minor modifications and additional statistics the manuscript is suitable for publication:

1.) The authors use a three tandem mNG, and achieve better localization precision (Fig. S3+4). The addition of 3 fluorophores will result in a tag of more than 75 kDa and possibly 12 nm length. There is the general trend in microscopy to reduce the tag size. Additional distance from the protein of interest,

leads to a degree of uncertainty about its actual position.
Please explain / comment on this caveat in the text.

2.) For their first experiments the authors study a yeast septin and rather long rounds of imaging. Septins have certainly a lower turnover than actin and microtubules, but there should be some movement of localizations during 100-200 sec. Were no displacements detectable or negligible? Or were they somehow compensated?
Please explain / comment on this caveat in the text.

3.) Fig. 3 is a bit misleading for the reader. These are independent measurements of two groups. The linear fit implies some kind of a course. A graph with: "before" and "after" bleach on the x-axis, and localizations on the y-Axis, and a line connecting both (before and after bleach) spots would be more intuitive.

4.) Line 333

Figure S6 shows the super resolution image reconstructed from data acquired for different amounts of time...

There is no image, just the graph for achieved resolution. However, images would be interesting to directly assess the performance of the method and the resolution increase after 1.5 sec of data acquisition.

5.) Figure S1 is this a single experiment / are statistics missing?

6.) Figure S3 are only 2 replicates. 3 independent experiments and statistics are eligible.

7.) For Figure S4 also statistics, informations on n numbers ... are eligible.

8.) In Fig S5 statistics are missing.

Reviewer #3 (Remarks to the Author):

This work describes LIVE-PAINT, a new method for labeling proteins in live cells, derived from the principles of DNA-PAINT and using the PAINT imaging method. LIVE-PAINT has the potential to expand our toolbox of available fluorescent labels for SMLM that is currently very restrictive for live imaging. The LIVE-PAINT method described by Oi et al. is functional and applicable.

There are some concerns about this manuscript. First, as explained below, some statements about this technique unsupported experimentally. Second, the manuscript is in a "draft" format with the need for major work on both the text and the figures.

Specific comments, major conceptual concerns:

1- There is no SMLM application of LIVE-PAINT to test the capability of the technique to resolve biological structures that are separated by less than ~250 nm, the limit of resolution of traditional light microscopy techniques. Measuring the septum splitting from a single hourglass ring to a doublet is readily performed by confocal and widefield microscopy (e.g. Lippincott et al. JCS, 2001) as the gap between the two rings is greater than the limit of resolution of conventional microscopy techniques (~250 nm). An appropriate test for the application of LIVE-PAINT to SMLM would be to resolve

structures that are separated by less than ~ 250 nm in live cells and this was not performed in this version of the manuscript. This is a major flaw of this work.

2- The principle of PAINT (Sharonov et al.) is based on having a large reservoir of unbound fluorescent molecules that can collide with its target, remain bound for a short period of time and then unbind or be photobleached. Careful optimization of imaging parameters (laser power density, exposure time and camera frame rate) were performed and described in both Sharonov et al. PNAS, 2006 (PAINT) and Jungmann et al. Nano Letters, 2010 (DNA-PAINT). In this work however, imaging parameters were not optimized possibly resulting in the suboptimal results with moderate expression of the fluorescently tagged construct. Additionally, modifying the fitting parameters of the localization algorithm could improve the data in cells expressing high levels of the construct. The lack of optimization of the acquisition and analysis parameters are major concerns about this work. In this work, the concentration of galactose was increased to drive increasing amounts of the fluorescently labeled binding protein, an important control. The measurement of fluorescent protein concentration at each galactose concentration would be very valuable information as it would enable comparison between expression method and expressed constructs. Measuring protein concentration in live yeast cells can be done using a fluorescence calibration curve (e.g. Wu and Pollard, Science, 2005 and Lawrimore et al. JCB, 2011).

Consequently, the following statement at line 85 is misleading based on the data shown in Figure 2. "By having a large reservoir of fluorophores that can exchange with the bleached ones, many localization events can thus be captured, enabling very high-resolution images to be collected. Localizations with low precision can be discarded, which also contributes to increased resolution.". Indeed, what appears to be a moderate expression of the fluorescent construct causes suboptimal datasets.

Multiple options are available to mitigate this problem but were not investigated:

- Can this issue be mitigated by the use of the tandem construct?
- Can this be addressed with more advanced localization algorithms that would eliminate lower quality emissions? In fact, can you map the precision value of each emission and determine the difference between the immobilized emission and the diffuse cytoplasmic emissions? Presumably the diffusing molecules have a lower precision value.
- Can you improve the data by varying the photon threshold to only select higher quality emissions thus eliminating the diffusing cytoplasmic emissions while keeping those specifically immobilized at the bud neck?
- Can you modify the localization parameters to add an "emission duration" cutoff as diffusing particles would likely be captured for a shorter period of time than immobilized particles?
- Importantly, can you run the camera at a faster frame rate? EMCCD cameras can reach faster frame rates if the field of view is restricted to a smaller frame.

Other important issues:

6- "Labeling using a construct with three tandem copies of mNG improves localization precision compared to a single copy".

What is the signal comparison between a single copy and three copies? Figure S4B shows a ratio but a graph of distribution of photon values for single and triple labeled construct would be much more informative.

What happens to the cytoplasmic noise? Figure 2 shows that only a small increase in expression of the single label construct results in a high level of background. What happens when the construct is triple tagged? Could you use the triple tagged construct expressed with 0.005% galactose and improve resolution?

What changes need to be made to the acquisition parameters when using a triple versus a single tag

construct?

8- "LIVE-PAINT enables longer data acquisition times". There are multiple concerns with this section.
A- The rationale for the design of this experiment isn't clear. Why use this approach (acquire, partial bleach, acquire) when you could simply acquire datasets for an extended period of time (~5 min) and then plotting the number of localizations versus time.

B- Show the datasets/images that were analyzed in the graph of Figure 3. What does the data look like?

C- If PAINT requires the fast binding and unbinding of the fluorescent label to the protein of interest, how can single emissions be captured for the directly labeled Cdc12 (data used for the black curve)? What do these single emissions look like?

5- The data in Figure S5A doesn't seem to show the stated effect. Are the data in each graph significantly different for the other? Superimposing the curves may show near perfect overlap.

7- Please provide a reasoning or reference for the following statement LINE 256 "This photobleaching reduces the resolution of the image because it limits the density of emitters that can be measured." Cumulative density does not necessarily result in higher resolution, especially in live SMLM.

10- Using LIVE-PAINT to label actin does not improve the currently used methods. LIVE-PAINT yields the exact same results as tagging actin directly with GFP. Actin tagged with GFP (or other tags as seen in Chen et al. *J Struct Biol*, 2012), is only incorporated into Arp2/3 polymerized actin patches and is excluded in all Formin polymerized actin filaments. Therefore LIVE-PAINT does not push the field forward in terms of tagging actin.

Two issues with the last full sentence in the legend of Figure 5 "Actin cables or rings are not observed either because we are imaging in TIRF or because the stringent structural requirements for actin in these structures means that even actin with very small ~2 kDa tags may be excluded from ring and cable structures²³". First, that statement should be in the text of the results section, not in a legend. Second, the authors would have seen cables and rings if they had been present because they can focus through the entire thickness of a thing budding yeast cell even with TIRF. Consider deleting that part or provide an explanation why TIRF would prevent seeing cables and rings.

11- LIVE-PAINT enables long tracking times in vivo. This data does not show the application of LIVE-PAINT to the measurements of protein dynamics using SMLM imaging technique in vivo. This work is done with diffraction limited microscopy, which is perfectly fine as it makes LIVE-PAINT a broadly applicable method, but it should be clearly stated in the text.

Why does the text not describe the data shown in the six different panels of Figure 6? This is beautiful analysis of particle diffusion but there's no description or comparison with known kinetics of Cofilin.

The statement "We observed a wide range of behaviors" needs to be substantiated with data. What range of behaviors? Why are they relevant and important to this work?

Does labeling cofilin affects its function? Were controls performed to measure cell viability and actin network dynamics? This data needs to be shown.

13- LINE 414. "...concurrent super-resolution imaging of multiple targets.". Showing the application of LIVE-PAINT with two spectrally distinct fluorescent proteins would increase the significance of this work for the field of SMLM. As the authors point out, SMLM imaging in live cells is restricted to a single color of fluorescence protein. New methods to expand the toolkit are needed. LIVE-PAINT could offer an option but without proof of applicability, option of co-expressing two fluorescent constructs at the

appropriate cellular concentrations, this remains only a hopeful thought.

14- LINE 452. "Two of the three peptide-pairs that we tested were suitable for LIVE-PAINT." Three techniques weren't shown. Wouldn't it be useful to describe the three tested options and explain why one didn't work?

15- Why were cells imaged in water? Is there no effect of keeping budding yeast cells in water rather than in medium? One would think that the change in the osmolarity would affect key molecular processes such as endocytosis.

Minor issues

3- Figure 1C. Add a line showing threshold photon value on the graph to identify the noise from the positive localizations.

16- Figure 2. Why is there cell wall/cell periphery labeling in the TRAP 0% image?

Response to reviewers

Here we respond to all reviewer comments, point by point. *Reviewer comments are in italics*, our responses are in plain text.

Reviewer #1 (Remarks to the Author):

Manuscript by Oi et al. reports on bipartite fluorescent protein labeling systems for live-cell imaging in yeast. The system relies on transient interaction of peptides, one attached to the protein of interest, the other one – to a fluorescent protein. The Authors claim the utility of the approach for super-resolution imaging. The Manuscript presents results on PAINT-like super-resolution in living yeasts with two interacting pairs (TRAP4-MEEVP and SYNZIP17-SYNZIP18) and one fluorescent protein (mNeonGreen).

The Referee finds the results promising.

The authors thank the reviewer for this positive comment.

The current version of the Manuscript lacks a recent scientific context and overstates the novelty of the reported results. The scientific presentation can be improved.

In light of the reviewer comments, we have extensively revised the scientific presentation, and we have added a reference to *Perfilov 2020*, DOI: 10.1007/s00018-019-03426-5 and to *Molle et al.* DOI: 10.1016/j.copbio.2015.12.009.

The Referee is mainly concerned with the choice of the self-blinking fluorescent protein as a reporter and with the high affinity of SYNZIP17-SYNZIP18 pair.

1) *The choice of the self-blinking fluorescent protein as a reporter*

mNeonGreen does indeed have the ability to blink intrinsically. We chose mNeonGreen because it is the brightest, monomeric FP currently available to us (Shaner et al. Nat. Methods, 2013). It is possible that some of the observed localization events could be a result of such blinking. We have added a sentence mentioning this point in the revised manuscript: “Although mNG is known to blink intrinsically, we chose to use it in our experiments because it is very bright and therefore can produce very precise localization events” (lines 137-139).

However, although there is likely a contribution from such blinking, the key effect that we are monitoring is due to ‘on/off’ binding of the peptide-binding-protein FP to the peptide that is fused to the protein of interest.

The key evidence supporting this statement is that our method works with different fluorescent proteins (mKO and mOrange). These fluorescent proteins are not known to blink intrinsically.

We have added an additional supplementary figure S1 (now referenced in lines 141-143) and two additional supplementary videos S1 and S2 which we have uploaded to Edinburgh DataShare with provided DOI: <https://doi.org/10.1101/2020.02.03.932228>.

Single molecule localisations with our method, but using mKO or mOrange as the FP, work in the same way as when we use mNeonGreen. Their behavior in our system is similar to what we observe when we use mNeonGreen, though the blinks are less bright, as would be expected because mKO and mOrange are not as bright as mNeonGreen. This is why we chose to use mNeonGreen in the experiments we present.

We thank the reviewer for providing critical feedback that has helped to improve the manuscript.

2) The choice of the high affinity SYNZIP17-SYNZIP18 pair.

The idea of the transient binding of diffusing probes being used in PAINT super-resolution imaging methods, we believe was first proposed by Robin Hochstrasser and colleagues in their 2006 PNAS paper, which we reference. We also mention protein-PAINT (Bozhanova et al. Chem Sci., 2017). We also thank the reviewer for bringing the review by Molle et al. to our attention. We have added a reference to the review by Molle et al. in line 70 of the revised manuscript, in the following sentence: “Unlike PALM and STORM methods, which are limited by photobleaching of the dye molecules over time, in PAINT-based methods there is continual replenishment of the fluorescent probes, which allows much longer imaging times, resulting in a higher density of localizations, and the potential for a higher resolution image” (Molle et al. Curr. Opin. Biotechnol., 2016).

Our method is distinguished by the genes that encode all the components being integrated into the chromosome. Thus, we have made strains that we can use over and over again, to obtain data on ideal biological replicates, and also to perform a variety of different measurements. Neither transient transfection nor exogenous addition of small molecule labels is required.

There are numerous reversible peptide-protein pairs available for use with our method. Many have already been validated, in published work, for use in yeast. For example Chen et al. ACS Synth Biol., 2015) In the current paper we show data for two such pairs (a TPR-peptide and a 2 stranded parallel coiled coil). Demonstration of the use of additional pairs is beyond the scope of this paper.

Nevertheless, to support our statement we have added additional data, presented as figures in the SI. These data show four different peptide-protein interaction pairs that are compatible with our method (Figure S2, now referenced in lines 150-151). The sentence reads: “We also provide evidence that LIVE-PAINT can be performed with additional peptide-protein interaction pairs (Figure S2)”.

There are many other interaction pairs we have not tested that would most likely be suitable as well (see Chen et al. ACS Synth Biol., 2015; Thompson et al. ACS Synth Biol., 2015). These data, combined with the data we provide demonstrating our ability to use our method with different fluorescent proteins provides strong support for our claim that our method can be easily extended to concurrent multicolor imaging in live yeast.

The Referee would like to share the following comments on the Manuscript.

Major

[1.1] novelty and scientific context

The authors have overlooked a recent demonstration of PAINT-like super-resolution and live-cell imaging with reversible interactions of small peptides (see Perfilov 2020, DOI: 10.1007/s00018-019-03426-5). Since the Manuscript under consideration is not the first one to show the applicability of transient peptides for live-cell PAINT-like super-resolution of intracellular proteins, the text should be amended accordingly.

The authors thank the reviewer for bringing this work to our attention. We have amended the discussion section of the manuscript accordingly: we have added a reference to Perfilov 2020, DOI: 10.1007/s00018-019-03426-5 and discuss their work in lines 425-427: “Recently, interacting charged coiled coil pairs have been used to label proteins to perform PALM imaging in live mammalian cells²⁹. This work provides a valuable independent validation of our approach”.

We thank the reviewers for pointing out this omission, which we have now remedied (lines 425-427). Although the approach of Perfilov and colleagues similarly takes advantage of coiled coil interactions for fluorescence imaging, the super-resolution performed in live cells was performed using PALM (with Dendra-2), and not PAINT. Their major finding fully supports one of our key conclusions - replenishment by

exchange means that imaging can be performed for longer periods of time, thus allowing a greater number of localisations to be observed.

The work by Perfilov et al. is important, and we are pleased that it provides independent evidence for the validity of our work.

[1.2] To put things into perspective, both the current Manuscript and the one by Perfilov are, in fact, live-cell adaptations of IRIS (image reconstruction by integrating exchangeable single-molecule localization, DOI: 10.1038/nmeth.3466). Other examples of small peptide-based tags for live-cell super-resolution imaging exists for nM affinity range (close to the one of SYNZIP17-SYNZIP18), e.g., DOI: 10.1038/ncomms10372. The super-resolution with the transient binding of diffusing probes is an ancient concept; see review by Molle et al. DOI: 10.1016/j.copbio.2015.12.009 The Referee oppose the introduction of yet another catchy abbreviation to the field (LIVE-PAINT). Besides the lack of novelty (see point 1.1), it does not convey the idea that the system is limited to protein labeling.

With regards to our selection of interaction pairs, please see our response to “2) The choice of the high affinity SYNZIP17-SYNZIP18 pair”.

Regarding the reviewer’s dislike of our newly coined abbreviation, we note that none of the other reviewers were of this opinion. It is convenient to have succinct ‘catchy abbreviations’ for referring to a method - see PALM, STORM, STED, PAINT, DNA-PAINT etc. We carefully chose this name to distinguish it from methods that adopt PAINT in their names connected with proteins. It is better that we, the authors, propose an easily memorable and reusable name, than people make up their own abbreviations for a long name.

[2] Support of some claims by the data may not be sufficient

[2.1] choice of mNeonGreen as a reporter fluorescent protein raises concerns due to its intrinsic blinking.

The authors insist that the observed blinking is a manifestation of transient binding. However, mNeonGreen is robustly blinking all by itself. See original paper by Shaner (DOI: 10.1038/nmeth.2413) and a more recent one (DOI: 10.1016/j.bbrc.2019.11.163). The Referee doubts that the PAINT-like localizations analyzed throughout the text reflect solely binding events and not blinking dynamics of mNeonGreen. Please discuss the impact of mNeonGreen blinking on the conclusions of the study.

Please refer to our response to “1): The choice of the self-blinking fluorescent protein as a reporter.”

[2.2] Claim of wide applicability and multi-color labeling

Is the method at all usable with non-blinking or non-photoactivatable FPs for super-resolution? What are other fluorescent proteins suitable?

Please refer to our response to “1): The choice of the self-blinking fluorescent protein as a reporter.”

Please support the claims:

LL421 "...is limited only by the number of orthogonal peptide-protein interaction pairs and the number of available spectrally distinct FPs, both of which are abundant."

Please refer to our response to “2) The choice of the high affinity SYNZIP17-SYNZIP18 pair.”

LL412-413 "Any FP can be used."

Please refer to our response to “1): The choice of the self-blinking fluorescent protein as a reporter.”

– The Referee believes that these claims are not supported by the data in the current version of the Manuscript.

[2.3] The choice of tight (nM-scale binding) peptide pair is not explained and may conflict with the notion of rapid exchange.

The authors start with:

LL122-124 "The peptide-protein interactions are chosen so that solution exchange occurs on a timescale shorter than or comparable to the bleaching lifetime."

But:

LL134 The peptide pair appears to bind quite tightly (~1nM), which is not in line with the idea of PAINT-like exchange. Please elaborate on the choice of peptide pair.

Please refer to our response to “2) The choice of the high affinity SYNZIP17-SYNZIP18 pair.”

As stated in the original version of the manuscript, we started with three different interaction pairs: (1) a charged coiled coil, which we dropped because using it gives rise to cells with abnormal morphology, (the K_d of this interaction pair is 5 nM), (2) a peptide-

TPR interaction (K_D 300 nM), and (3) the SYNZIP17/18 coiled coil interaction pair (K_D 1 nM).

We continued our work with interaction pairs (2) and (3), specifically to investigate how differences in K_D influenced the method. A comparison of data collected with interaction pairs (2) and (3) is shown in Figure 3. We have also added more details about (1), including our rationale for selecting the peptide-protein pairs we use.

Our rationale reads: “TRAP4-MEEVF¹⁵, SYNZIP17-SYNZIP18¹⁷, and CC-A_N^{3.5}-CC-B_N^{3.5}³³ interaction pairs were identified in the literature and tested *in vivo* in live yeast by fusing one half of the interaction pair to a protein of interest and the other half to a FP. The cells were imaged under a microscope as described in the “Microscopy” section. The TRAP4-MEEVF and SYNZIP17-SYNZIP18 interaction pairs showed no morphological abnormalities, no noticeable growth defect in liquid culture or on plates and showed fluorescence at the expected locus of the tagged protein. For this reason, these two interaction pairs were used in this work. Using CC-A_N^{3.5}-CC-B_N^{3.5} interaction pair resulted in unusual cell morphology in the vast majority of cells, with elongated cell shapes, so this pair was not used further. The 101A-101B and 108A-108B interaction pairs¹⁸ were also checked and did not cause any changes to cell morphology or changes to cell growth.” (lines 512-522)

We used two different interaction pairs, with K_D 300 nM and K_D 1 nM, and showed that either can be used with our method. It is important to note that it is the on/off rates, rather than solely the K_D , that are important for exchange based methods.

Extensive *in vivo* quantification is not available for any system, nevertheless there are some numbers, measured *in vitro*, that relate to our choice of peptide-protein pairs.

DNA-PAINT typically uses imager docking strand pairings that are 9 or 10 nucleotides long, which have off rates (k_{off}) of the order of 2 s^{-1} and 0.1 s^{-1} , respectively. And both have k_{on} of the order of $10^6 \text{ M}^{-1} \text{ s}^{-1}$ (Jungmann et al. Nano Lett., 2010)

k_{off} rates for a series of coiled coils (not the exact sequences we are using) have been reported to be of the order 0.01 to 0.1 s^{-1} (Groth et al. Chem Sci., 2018). From that paper, they report for different pairs, with different K_D s: $K_D = 4 \times 10^{-6} \text{ M}$, $k_{off} = 0.04 \text{ s}^{-1}$; $K_D = 1 \times 10^{-9} \text{ M}$, $k_{off} = 0.07 \text{ s}^{-1}$; $K_D = 5 \times 10^{-10} \text{ M}$, $k_{off} = 0.01 \text{ s}^{-1}$

k_{off} measured for peptide-TPR interaction is of the order 0.7 s^{-1} (Jackrel et al. Protein Sci., 2009)

These values are consistent with our choosing peptide-protein pairs that have appropriate binding characteristics to be used in LIVE-PAINT experiments.

Even more importantly, our data using a peptide-TPR pair and a coiled coil pair, and the pGAL1 promoter to vary expression levels, show that our choices are appropriate for applying LIVE-PAINT within live yeast cells. We will further extend and refine those choices as we develop the technology, showing how it can be optimized to image different proteins.

[2.4] Evidence of exchange is insufficient

The experimental evidence does not convince the Referee of the exchange in the system. Moreover, the Referee challenges the idea that ~1nM Kd allows for such exchange at the imaging timescale.

See Figure 1 (mNeonGreen) and also Figure S1 (mKO and mOrange), which shows blinking events in individual frames, demonstrating that fluorescent proteins which do not intrinsically blink due to their photophysical characteristics, exhibit blinking behavior with our method (due to binding/unbinding of the peptide-binding-protein FP).

Specifically, the photostability of the mNeonGreen-labeled Cdc12 (conventional fuse) should be assessed in direct comparison (Figure 4). Otherwise, the difference between the curves can be attributed to sampling. See also point [3.6]

The purpose of Figure 4 is to demonstrate that when we express more fluorescent proteins, by increasing the galactose concentration in the media and expressing the peptide-binding-protein FP from the pGAL1 promoter, the rate of observed localization events does not decay as quickly as it does for low concentrations of galactose. This means that we can simply express more fluorescent proteins and image for longer to achieve more localization events.

Comparing to a direct fusion of the protein-of-interest to mNeonGreen is not appropriate here, since the natural abundance of different proteins-of-interest varies tremendously inside the cell. This means that dramatically different results would be obtained, depending on the protein of interest we would choose to use as an example. For this reason, a direct fusion to mNeonGreen would not be a meaningful comparison to the other curves shown in Figure 4.

Using a direct fusion would not provide any information to support or refute our point that increasing the number of fluorescent proteins decreases the decay rate in localization rate.

We are grateful to the reviewer for motivating us to revisit the data in Figure 4 and S5 from the original manuscript. Prompted by these comments, we reanalysed the data and identified a bug in our data processing script. Specifically, we previously plotted one minus the cumulative probability distribution for localizations, as a function of time. This is not technically correct, because it implies that the localization rate goes to zero at the end of our video, even when it does not. Instead, we have now binned and summed the localizations, to get a number of localizations for each “time chunk” of a video. We then normalized these values to the highest value for each condition.

We have therefore corrected our data processing script, and we have now combined Figure 4 and (along with Figure S5 from the original manuscript) into a new Figure 4, to better present these data. Figure 4 shows that the data for different concentrations of galactose are clearly not the same. We see that as we increase the concentration of galactose (so that more peptide-binding-protein FP is expressed and consequently there is a larger cytoplasmic reservoir for exchange) imaging can be continued for longer times. Quantitatively, we see that the exponential time constant (τ) varies from 1.9 s at 0% galactose to 139 s at 0.1% galactose, for the SYNZIP17-SYNZIP18 interaction pair. We also add an additional sentence to the text referencing Figure 4: “In Figure 4B, for example, we observe that when imaging Cdc12-SYNZIP18 + SYNZIP17-mNG using 0.1% galactose, even after 200 s of imaging, localizations are still being recorded at about 30-40% of the initial rate” (lines 326-329).

Figure 4. Localization rate decays more slowly with increased FP expression. Localization rate as a function of imaging time for (A) Cdc12-MEEVF + TRAP4-mNG and (B) Cdc12-SYNZIP18 + SYNZIP17-mNG, each at four different concentrations of galactose. Data for the MEEVF-TRAP4 interaction pair is for 0% galactose (red), 0.005% galactose (dark orange), 0.02% galactose (light orange) and 0.1% galactose (yellow). Data for the SYNZIP18-SYNZIP17 interaction pair is for 0% galactose (bright blue), 0.005% galactose (blue), 0.02% galactose (teal) and 0.1% galactose (mint). The data for each concentration of galactose were fit to a single exponential (shown as a solid line with matching color). For the MEEVF-TRAP4 interaction pair (A), the exponential time constant (τ) for the different concentrations of galactose is 0%: 4.7 s; 0.005%: 15 s; 0.02%: 32 s; 0.1%: 81 s. For the SYNZIP18-SYNZIP17 interaction pair (B), the exponential time constant (τ) for the different concentrations of galactose is: 0%: 1.9 s; 0.005%: 54 s; 0.02%: 73 s; 0.1%: 139 s. (lines 332-344).

Concerning the normalized y-axis of the plot, were the initial localization counts similar?

The initial localization counts were lower for lower concentrations of galactose, because the expression level of the fluorescent protein constructs scales approximately linearly with galactose concentration, because we are using the pGAL1 promoter with a GAL2 deletion (Hawkins et al. JBC, 2006). Also see SI figure S5 in the revised manuscript, (Figure S1 in the original manuscript).

The Referee is also a bit surprised by seeing both lambda and tau for the decay.

The authors thank the reviewer for noting that both lambda and tau are mentioned in the text for the decay. The reviewer is correct that because these values are simply the reciprocal of each other, it is inappropriate to mention both. We have amended the text to only report tau values. The new Figure 4 legend now reports the following: "For the MEEVF-TRAP4 interaction pair (A), the exponential time constant (τ) for the different concentrations of galactose is 0%: 4.7 s; 0.005%: 15 s; 0.02%: 32 s; 0.1%: 81 s. For the SYNZIP18-SYNZIP17 interaction pair (B), the exponential time constant (τ) for the different concentrations of galactose is: 0%: 1.9 s; 0.005%: 54 s; 0.02%: 73 s; 0.1%: 139 s" (lines 340-344).

Minor:

[3.1] Scientific presentation.

The abstract should be rephrased in neutral language. The Referee agreed to review this Manuscript after examining the abstract and was disappointed that none of the points in the last sentence was experimentally supported in the current version of the Manuscript.

Consider:

"widely applicable, easily implemented, and the modifications minimally perturbing, but it also allows extended data acquisition times compared to previously possible with methods that involve direct fusion to a fluorescent protein."

Either provide support for, or

Amend the text and abstract to reflect only the results reported in the paper.

In response to this critique, we chose to provide support for these statements.

"Widely applicable."

We show data for 3 different cellular proteins. The method is straightforward to implement. There is no reason to suppose that it is not widely applicable. We anticipate that when the method is published many researchers will adopt it, and its wide applicability will be demonstrated.

"Minimally perturbing."

We observe no changes in morphology or growth rate as a consequence of directly fusing a peptide to the C-terminus of the only copy of a gene. This is true for the proteins studied in this paper. It is of special importance to note that both actin and cofilin (proteins that are notoriously susceptible to functional perturbation if directly fused to a fluorescent protein) can be labelled using our method.

Finally, in a previous paper, we showed that fusion of a peptide to a membrane protein (Pma1) did not perturb function, whereas direct fusion to a fluorescent protein does. We have added a reference to this paper (Hinrichsen et al. PEDS, 2017) in lines 143-144. We did not reference in the abstract, because typically one does not put references in the abstract.

"...data acquisition times compared to... fusion to a fluorescent protein."

We have changed the text to read "but we also anticipate it will extend data acquisition times compared to those previously possible with methods that involve direct fusion to a fluorescent protein." (lines 41-43)

[3.2] The Referee asks the authors to correct the following inaccuracies in the discussion:

LL391-394 *The point of 'main advantage over current small molecule-based PAINT' is not clear. To the best of the Referee's knowledge, protein-PAINT (Ref. 25, DOI: 10.1039/C7SC01628J) works within living cells. Please amend the text accordingly.*

We have amended the statement in the discussion to say “The main advantage over DNA-PAINT is that LIVE-PAINT works inside living cells; all the components that we describe are chosen to function in that *milieu*.” (lines 422-423 in the revised manuscript.)

We address the advantages of LIVE-PAINT over methods such as protein-PAINT in the subsequent paragraph: “Other approaches to performing PAINT in live cells, such as protein-PAINT, require the addition of organic dyes, cannot be used to image multiple targets simultaneously, and require a larger fusion to the target protein³¹” (lines 438-441).

[3.3] *Please discuss the potential limitations and narrow the area of applicability of the method. The obvious point is the impact of the background fluorescence on the SNR.*

Our method has a higher background compared to PALM, but this effect does not significantly narrow the area of applicability of our method. In fact, as we note, our method has a greater area of applicability than PALM, because of its potential to image both very high and very low abundance proteins in live cells, and also because of the potential to extend our method to multicolor imaging.

As is clear from the data we present that our method has a suitably high SNR to be generally useful. Moreover, the SNR can be optimized by varying the interaction pair and expression level of the peptide-binding-protein FP. (see Figures 1 and 2). SNR can also be increased by TIRFM (which we use) and by light sheet fluorescence microscopy (we have added a comment saying LSFM could be used): “Unbound FPs in LIVE-PAINT result in background fluorescence, but this effect can be mitigated by reducing the illumination volume in the cell, as we have done using TIRFM in this work, or by using other strategies, such as light-sheet fluorescence microscopy (LSFM)” (lines 477-480).

[3.4] *LL140 "Both TRAP4-MEEVF and SYNZIP17-SYNZIP18, were well-tolerated by the cell." It is common to support these types of statements ("well-tolerated") quantitatively. Consider including cell viability, imaging, or other data in support of this claim.*

We observed no morphological differences in the yeast strains containing the TRAP4-MEEVF and SYNZIP17-SYNZIP18 interaction pairs. By contrast, when Cdc12 is directly

fused to a fluorescent protein, ~5% of the cells show a distorted morphology (Hinrichsen et al. *PEDS* 2017). In addition to the cited work, in these studies we also observed a small fraction of cells with distorted morphology when Cdc12 was directly fused to a fluorescent protein. We observed no changes in growth rates of liquid cultures of cells expressing the TRAP4-MEEVF and SYNZIP17-SYNZIP18 interaction pairs, compared to the parent strain.

The statement ‘well-tolerated’ was used for brevity, and intended to cover all these points, but we have revised the manuscript to state explicitly what we mean: “We observed no distorted cell morphology or changes in growth rate in liquid media when using the TRAP4-MEEVF and SYNZIP17-SYNZIP18 interaction pairs. In previous work we observed distorted cell morphology for ~5% of yeast expressing a direct fusion of Cdc12 to an FP²³” (lines 147-150).

[3.5] *The Referee asks the authors to consider and discuss the possibility of self-dimerization of tested coiled-coils, peptides, or their non-specific interaction with cellular proteins. Provide experimental support for the lack of non-specific interactions, if possible, or an appropriate citation.*

When we tag different proteins we do not see any “incorrect” structures (e.g. no bud neck septum when we tag actin or cofilin and no actin/cofilin puncta when tagging Cdc12). If there are off-target interactions with other cellular proteins, they are so weak that they do not appear when we image. If the coiled coils we used exhibited any significant self-dimerization, we would not be able to use them in our imaging method. We carefully chose a coiled coil pair for which no significant self-dimerization has been reported, see Thompson et al. *ACS Synth Biol.*, 2012, which presents extensive characterization of the SYNZIP17-SYNZIP18 interaction pair

We have provided an additional supplementary figure (S3, now referenced in lines 150-154) which demonstrates labeling specificity *in vivo* for the TRAP4-MEEVF and SYNZIP17-SYNZIP18 interaction pairs.

For future imaging experiments where we use multiple interaction pairs at once in the cell, we plan to use the 101-108A/B interaction pairs (supplementary figure - S2 in the revised text (now referenced in lines 150-154) - shows imaging data using 101A/B and 108A/B). These interaction pairs have been shown to be orthogonal to each other and have been used to specifically label cellular structures in live yeast (Chen et al. *ACS Synth Biol.*, 2015). See also Thompson et al. *ACS Synth Biol.*, 2012, which provides evidence that the SYNZIP17-SYNZIP18 interaction pair we use does not self-dimerise.

[3.6] *The Referee likes the idea of Figure 3 very much. However, in its current form, the figure is neither convincing (in comparison with classic FRAP experiments), nor quantitative (linear fit for 'ease of visualization' does not provide a quantitative measure of the change in apparent photostability). Moreover, since the DBSCAN filtering of 'specific' vs. 'non-specific' events was applied to the time series, it would be helpful to see maximum projections of the plotted time slots to convince the reader that only specific localizations are counted in such a plot. The Referee thinks that it would be a great Figure, once additional replicas and quantitative analysis are included.*

We thank the reviewer for their optimism about the content of Figure 3. We have taken their advice and included maximum projections for time ranges through the course of the experiment and added them to our Figure 3. We have also reformatted the raw data displayed in the figure to better show the reduced bleaching observed with our reversible interaction pairs compared to the direct fusion to a fluorescent protein. These changes are reflected in the updated Figure 3 in the revised manuscript.

The new Figure 3 legend text now reads: “Figure 3. LIVE-PAINT shows recovery of signal after bleaching. (A) LIVE-PAINT interaction pairs show more recovery in number of localization events than a direct fusion to a FP. In this experiment, fluorescence images were collected for 1,000 frames (50 s) at standard imaging power (3.1 W/cm²), then the sample was photobleached using high laser power (26.6 W/cm²), and then the sample was again imaged for 1,000 frames (50 s) at standard imaging power. Cdc12-SYNZIP18 + SYNZIP17-mNG (blue/green circles, each representing a single cell) retain many more localization events than Cdc12-mNG (gray circles, each representing a single cell) after two minutes of photobleaching. Each shade of gray or blue/green represents a single cell, which can be color-matched between pre-photobleaching (PrB) and post-photobleaching (PoB) conditions. DF = Cdc12-mNG (Direct Fusion); SZ =

Cdc12-SYNZIP18 + SYNZIP17-mNG (SYNZIP pair). (B) Maximum projections for different frame ranges in both “before bleaching” and “after bleaching” videos demonstrate that signal obtained after bleaching continues to localize to the yeast septum. (Top) Maximum projections are shown for 200 frame ranges for a representative cell expressing Cdc12-mNG. (Bottom) Maximum projections are shown for a representative cell expressing Cdc12-SYNZIP18 + SYNZIP17-mNG. All “before bleaching” images are normalized to one another and, similarly, all “after bleaching” images are normalized to one another. Scale bar is 1 μm .” (lines 293-310)

We have also supplied a supplementary figure (Figure S9, now referenced in lines 290-291), which shows maximum projection images of each of our analyzed cells in the ‘before’ and ‘after’ bleaching conditions. We thank the reviewer for the constructive feedback on this figure.

[3.7] *The exact number of independent samples should be stated in figure legends. In cases where the representative image or graph is provided (such as Figure 1C), the exact number of experiments with similar results should be indicated.*

Where appropriate we supply information about the number of samples or cells analyzed. Figure 1C, however, is a representative trace to demonstrate the idea of PAINT exchange, of which there will be many for each experiment performed. We saw similar traces for all our experiments.

[3.8] *The Referee has failed to find Supplementary Movies 1 and 2 within the submission files*

The authors apologize for this oversight. We have uploaded supplementary movies 3 and 4 (referenced as supplementary movies 1 and 2 in unrevised text, now listed as 3 and 4 due to the addition of two new supplementary movies) to the Edinburgh DataShare and the movies can be viewed at the following DOI: <https://doi.org/10.1101/2020.02.03.932228>.

[3.9] *Please include exact measurements and calculations behind the power density values ($\sim \text{W}/\text{cm}^2$) included in the paper. These values are essential for the design of imaging experiments, yet the literature in the field contains numerous examples of order-of-magnitude errors, uncorrected for years. Preferably, include the power measurements at the sample plane without TIRF, calculated or measured irradiation area (it could be measured by bleaching and then shifting the sample a bit). The Referee expects a bit higher values from Cobolt MLD laser.*

We have added the following extra information in the methods section: “The lasers were first attenuated with neutral density filters to reduce the excitation power. The power at the back aperture of the objective lens was measured, and the excitation area determined using tetraspeck beads immobilised on a glass coverslip” (lines 607-610).

[4] *Data availability and independent analysis. Please provide raw time series behind the graphs and images (Figures 1, 2, 4, 5, 6) on Figshare.*

[4.1] *The Referee was not able to provide comments on single-molecule analysis without [4]*

Once our paper is in press, we will deposit in Edinburgh DataShare, a digital repository of research data produced at the University of Edinburgh, hosted by Information Services and obtain a DOI, which will be included in the paper. Similarly, a citation to the paper will be included along with the DOI. This is the recommended route for data deposition.

Reviewer #2 (Remarks to the Author):

The manuscript by Oi et al. describes a new approach for superresolution and single molecule microscopy. The authors use a variation of PAINT (LIVE-PAINT), which is renowned for its high sampling rates. The complementary oligos of DNA-PAINT are exchanged by small interacting peptides. These peptides can be expressed in live cells and are, similar to DNA-PAINT, replenished for a theoretically unlimited number of imaging rounds. The authors use tetratricopeptide repeat affinity proteins (TRAPs), which were used by the lab for live imaging before and synthetic peptides derived from bZIP leucine zippers (Grigoryan et al. 2009).

The authors use yeast for their experiments and perform live imaging of septin, actin and cofilin. They provide evidence that their LIVE-PAINT is suitable for Live-SMLM of proteins with a slow to fast turn-over.

Although the presented yeast images might not be spectacular at first glance, the method has great potential for a multitude of applications. The modification and adaption of this method to other cell types (e.g. mammalian cells) will be highly relevant for many researchers.

With minor modifications and additional statistics the manuscript is suitable for publication:

We thank the reviewer for their thoughtful comments, and appreciate this final summary that with minor modifications the paper is suitable for publication.

1.) *The authors use a three tandem mNG, and achieve better localization precision (Fig. S3+4). The addition of 3 fluorophores will result in a tag of more than 75 kDa and possibly 12 nm length. There is the general trend in microscopy to reduce the tag size. Additional distance from the protein of interest, leads to a degree of uncertainty about its actual position.*

Please explain / comment on this caveat in the text.

The reviewer is correct and we have added a comment on this caveat in the text: “We note, however, that the larger size of the three tandem mNG construct creates additional distance between the protein-of-interest and therefore leads to increased uncertainty about its actual position.” (lines 262-264).

2.) *For their first experiments the authors study a yeast septin and rather long rounds of imaging. Septins have certainly a lower turnover than actin and microtubules, but there should be some movement of localizations during 100-200 sec. Were no displacements detectable or negligible? Or were they somehow compensated?*

Please explain / comment on this caveat in the text.

We did not observe any visible change in the position of the localization events observed during the imaging rounds. However, it is worth noting that due to the reversible nature of the interactions used in LIVE-PAINT, it would be difficult to distinguish between the protein of interest moving and a new binding or unbinding event. We do not suggest that no protein-of-interest moved during the course of the experiments. However, the overall shape of the yeast septin structure did not change noticeably during the course of an imaging experiment.

3.) *Fig. 3 is a bit misleading for the reader. These are independent measurements of two groups. The linear fit implies some kind of a course. A graph with: “before” and “after” bleach on the x-axis, and localizations on the y-Axis, and a line connecting both (before and after bleach) spots would be more intuitive.*

The authors appreciate the feedback on the data formatting for Figure 3. We have supplemented the figure with maximum projection images for time ranges through the course of the experiment to provide evidence that the localizations obtained after bleaching continue to show specificity to the yeast septum. We have also reformatted the raw data displayed in the figure to better show the reduced bleaching observed with our reversible interaction pairs compared to the direct fusion to a fluorescent protein. These changes are reflected in the updated Figure 3 in the paper.

The new Figure 3 legend text now reads: “Figure 3. LIVE-PAINT shows recovery of signal after bleaching. (A) LIVE-PAINT interaction pairs show more recovery in number of localization events than a direct fusion to a FP. In this experiment, fluorescence images were collected for 1,000 frames (50 s) at standard imaging power (3.1 W/cm^2), then the sample was photobleached using high laser power (26.6 W/cm^2), and then the sample was again imaged for 1,000 frames (50 s) at standard imaging power. Cdc12-SYNZIP18 + SYNZIP17-mNG (blue/green circles, each representing a single cell) retain many more localization events than Cdc12-mNG (gray circles, each representing a single cell) after two minutes of photobleaching. Each shade of gray or blue/green represents a single cell, which can be color-matched between pre-photobleaching (PrB) and post-photobleaching (PoB) conditions. DF = Cdc12-mNG (Direct Fusion); SZ = Cdc12-SYNZIP18 + SYNZIP17-mNG (SYNZIP pair). (B) Maximum projections for different frame ranges in both “before bleaching” and “after bleaching” videos demonstrate that signal obtained after bleaching continues to localize to the yeast septum. (Top) Maximum projections are shown for 200 frame ranges for a representative cell expressing Cdc12-mNG. (Bottom) Maximum projections are shown for a representative cell expressing Cdc12-SYNZIP18 + SYNZIP17-mNG. All “before bleaching” images are normalized to one another and, similarly, all “after bleaching” images are normalized to one another. Scale bar is $1 \mu\text{m}$.” (lines 293-310)

While we have chosen not to include the figure with lines connecting data points “before” and “after” bleaching in the manuscript, we have plotted the data in that way and attach the figure here.

4.) *Line 333*

Figure S6 shows the super resolution image reconstructed from data acquired for different amounts of time...

There is no image, just the graph for achieved resolution. However, images would be interesting to directly assess the performance of the method and the resolution increase after 1.5 sec of data acquisition.

The authors appreciate this feedback. We have removed reference of these “images” in the manuscript and instead show only the boxplots. The images look nearly identical and we report that the resolution does not continue to improve after 1.5 s of acquisition and we therefore have chosen to show only the boxplots.

5.) *Figure S1 is this a single experiment / are statistics missing?*

Figure S1 (now figure S5 in revised text) is a single experiment. This experiment was performed to confirm the behavior of the pGAL1 reporter in the presence of a GAL2 deletion. The galactose dependence of the transcriptional response of the pGAL1 promoter was characterized in detail by the researchers who described this system (Hawkins et al. JBC, 2006).

6.) *Figure S3 are only 2 replicates. 3 independent experiments and statistics are eligible.*

The reviewer is correct that Figure S3 (now figure S7 in revised text) contains only two replicates. However, we are not making claims of statistical significance in this figure. This is the reason why we have provided the raw data for both replicates in the figure, in addition to the average of the two replicates.

7.) For Figure S4 also statistics, informations on n numbers ... are eligible.

Please see our response to point 6 above.

8.) In Fig S5 statistics are missing.

The data in Figure S5 (now show in Figure 4 in revised text) are each the result of a single experiment. We have reanalyzed the data and remade this figure, which is now given as Figure 4 in the revised manuscript. This figure now shows more pronounced differences between the different curves. Specifically, we observe that the rate of acquisition of localization events decrease significantly more slowly as we increase galactose concentration.

Figure 4. Localization rate decays more slowly with increased FP expression. Localization rate as a function of imaging time for (A) Cdc12-MEEVF + TRAP4-mNG and (B) Cdc12-SYNZIP18 + SYNZIP17-mNG, each at four different concentrations of galactose. Data for the MEEVF-TRAP4 interaction pair is for 0% galactose (red),

0.005% galactose (dark orange), 0.02% galactose (light orange) and 0.1% galactose (yellow). Data for the SYNZIP18-SYNZIP17 interaction pair is for 0% galactose (bright blue), 0.005% galactose (blue), 0.02% galactose (teal) and 0.1% galactose (mint). The data for each concentration of galactose were fit to a single exponential (shown as a solid line with matching color). For the MEEVF-TRAP4 interaction pair (A), the exponential time constant (τ) for the different concentrations of galactose is 0%: 4.7 s; 0.005%: 15 s; 0.02%: 32 s; 0.1%: 81 s. For the SYNZIP18-SYNZIP17 interaction pair (B), the exponential time constant (τ) for the different concentrations of galactose is: 0%: 1.9 s; 0.005%: 54 s; 0.02%: 73 s; 0.1%: 139 s. (lines 332-344).

Reviewer #3 (Remarks to the Author):

This work describes LIVE-PAINT, a new method for labeling proteins in live cells, derived from the principles of DNA-PAINT and using the PAINT imaging method. LIVE-PAINT has the potential to expand our toolbox of available fluorescent labels for SMLM that is currently very restrictive for live imaging. The LIVE-PAINT method described by Oi et al. is functional and applicable.

We are pleased that the reviewer recognizes the potential of our method and considers it functional and applicable.

There are some concerns about this manuscript. First, as explained below, some statements about this technique unsupported experimentally. Second, the manuscript is in a "draft" format with the need for major work on both the text and the figures.

Specific comments, major conceptual concerns:

1- There is no SMLM application of LIVE-PAINT to test the capability of the technique to resolve biological structures that are separated by less than ~250 nm, the limit of resolution of traditional light microscopy techniques. Measuring the septum splitting from a single hourglass ring to a doublet is readily performed by confocal and widefield microscopy (e.g. Lippincott et al. JCS, 2001) as the gap between the two rings is greater than the limit of resolution of conventional microscopy techniques (~250 nm). An appropriate test for the application of LIVE-PAINT to SMLM would be to resolve structures that are separated by less than ~250 nm in live cells and this was not performed in this version of the manuscript. This is a major flaw of this work.

In this work we did not aim to optimize the resolution. That aim is for our future work. In this work, we aimed to demonstrate the feasibility of the method, and to measure and

report resolution using the field standard Fourier ring correlation method (<https://doi.org/10.1016/j.jsb.2013.05.004>).

Regarding the specific comment, it is important to note that we were not measuring the distance between the septum rings, rather, we were measuring the total width of the septum. During early division, when there is only one ring, the septum width is on the order of 100-200 nm (Figure S2 in original manuscript, Figure S6 in revised text), which is less than the ~250 nm resolution limit of conventional microscopy techniques.

2- The principle of PAINT (Sharonov et al.) is based on having a large reservoir of unbound fluorescent molecules that can collide with its target, remain bound for a short period of time and then unbind or be photobleached. Careful optimization of imaging parameters (laser power density, exposure time and camera frame rate) were performed and described in both Sharonov et al. PNAS, 2006 (PAINT) and Jungmann et al. Nano Letters, 2010 (DNA-PAINT). In this work however, imaging parameters were not optimized possibly resulting in the suboptimal results with moderate expression of the fluorescently tagged construct. Additionally, modifying the fitting parameters of the localization algorithm could improve the data in cells expressing high levels of the construct. The lack of optimization of the acquisition and analysis parameters are major concerns about this work.

The papers that the reviewer cites: Sharonov et al. PNAS, 2006 (PAINT) and Jungmann et al. Nano Letters, 2010 (DNA-PAINT) are indeed our inspiration, and will guide future work. Our intention with this first paper was to demonstrate that the concept works in live cells, to explain how we have set up the system, and to present some data on how changing parameters affects the results. We believe that the current paper accomplishes all of this. We completely agree that our exploration of parameters was not exhaustive, but such work will be the subject of follow-up papers.

In this work, the concentration of galactose was increased to drive increasing amounts of the fluorescently labeled binding protein, an important control. The measurement of fluorescent protein concentration at each galactose concentration would be very valuable information as it would enable comparison between expression method and expressed constructs. Measuring protein concentration in live yeast cells can be done using a fluorescence calibration curve (e.g. Wu and Pollard, Science, 2005 and Lawrimore et al. JCB, 2011).

We are very familiar with the papers the reviewer cites concerning quantification of protein concentrations within yeast cells. In this work, we do not need to know the absolute concentrations. We need to be able to change concentration over a fairly wide

range - which we are able to, as detailed in Supplementary figure S5 (previously S1 in unrevised text).

Consequently, the following statement at line 85 is misleading based on the data shown in Figure 2. "By having a large reservoir of fluorophores that can exchange with the bleached ones, many localization events can thus be captured, enabling very high-resolution images to be collected. Localizations with low precision can be discarded, which also contributes to increased resolution."

Indeed, what appears to be a moderate expression of the fluorescent construct causes suboptimal datasets.

The data shown in Figure 2 demonstrates that there is a tradeoff between an increased reservoir of fluorescent proteins (which can improve imaging by providing more fluorescent molecules capable of producing localization events) and increased background. Our work demonstrates that this tradeoff occurs, but that conditions exist which produce a high ratio of specific localizations while producing relatively low background. Choosing the most appropriate interaction pairs, and the optimal peptide-binding-protein FP expression levels to use when imaging protein targets of different natural abundance will be a topic for future studies.

Multiple options are available to mitigate this problem but were not investigated:

- Can this issue be mitigated by the use of the tandem construct?

We show in Figure 2 that we can identify the conditions that give optimal SNR. The use of the tandem construct would not increase the SNR, because the bound and unbound molecules will both be brighter.

- Can this be addressed with more advanced localization algorithms that would eliminate lower quality emissions? In fact, can you map the precision value of each emission and determine the difference between the immobilized emission and the diffuse cytoplasmic emissions?

Presumably the diffusing molecules have a lower precision value.

We investigated this option and we do indeed use precision cutoffs in our analysis, and we remove low precision localizations. This approach does help to remove low quality localization events, which are more likely to be nonspecific. We have provided an additional supplementary figure (Figure S12, with new text referencing this figure at lines 623-625) to demonstrate the effect of varying both our precision and minimum photons per localization thresholds.

- Can you improve the data by varying the photon threshold to only select higher quality emissions thus eliminating the diffusing cytoplasmic emissions while keeping those specifically immobilized at the bud neck?

As mentioned in our response to the previous point, we also use a minimum photon per localization event threshold. As can be seen in our additional supplementary figure (Figure S12, with new text referencing this figure at lines 623-625), varying the precision cutoff results in the more dramatic removal of background localization events.

We applied both precision and minimum photon threshold cutoffs in the analysis performed and images presented in the manuscript.

- Can you modify the localization parameters to add an “emission duration” cutoff as diffusing particles would likely be captured for a shorter period of time than immobilized particles?

We have tried varying the “shift factor” variable in the ImageJ GDSC SMLM plugin, but we did not observe noticeable improvements to the images by varying this parameter. Below is shown data processed for one cell expressing Cdc12-SYNZIP18 + SYNZIP17-mNeonGreen, using different shift factors (scale bar = 1 micron), keeping all other image processing parameters the same.

- Importantly, can you run the camera at a faster frame rate? EMCCD cameras can reach faster frame rates if the field of view is restricted to a smaller frame.

Although it is possible to run at a faster frame rate, there is no advantage to doing so. The localisation precision is dependent on the number of photons collected. The time it takes to collect the optimum number of photons is dependent on the photobleaching lifetime of the fluorescent protein, which should match, as closely as possible, the time it remains bound to its target protein. We use a low excitation power to reduce photodamage, it is this time that limits how fast we can image, and not the frame rate of the camera.

Other important issues:

6- *“Labeling using a construct with three tandem copies of mNG improves localization precision compared to a single copy”.*

What is the signal comparison between a single copy and three copies?

This question is difficult to answer quantitatively. For example, we are not sure whether the 3x tag has increased or decreased stability in the cell, or how well it folds compared to its 1x counterpart, so we do not know how the copy number of the protein varies between these constructs.

We can say that we obtain a similar number of localization events per cell for the 1x and 3x constructs, which supports the idea that the copy number of the 1x and 3x constructs is similar when expressed using the pGAL1 promoter and an identical concentration of galactose. However, we do not have data which provides a direct, quantitative comparison between the absolute signal for the 1x and 3x constructs. We believe, however, that the data we present supports our claim that the quality of the localization events is higher when using the 3x construct.

Figure S4B shows a ratio but a graph of distribution of photon values for single and triple labeled construct would be much more informative.

We wanted to keep this a clean and interpretable plot. We chose to highlight the key differences between the single and triple FP constructs. We have therefore not changed this plot. It is now Figure S8 in the revised manuscript.

What happens to the cytoplasmic noise?

See answer to point 6 above. We observe no noticeable difference to the cytoplasmic noise. This is to be expected, since the unbound and bound tandem 3x construct are equally bright, just as the unbound and bound single fluorescent protein are equally bright.

Figure 2 shows that only a small increase in expression of the single label construct results in a high level of background. What happens when the construct is triple tagged?

We only tested the triple construct at one expression level. The relevant comparison for our work is between the triple tagged tandem construct and the single fluorescent protein construct, at the same expression level.

Could you use the triple tagged construct expressed with 0.005% galactose and improve resolution?

These data were obtained at 0.005% galactose. Due to the improved quality of the super-resolution localizations while using the triple tagged construct, we imagine that it can improve the resolution of the images compared to the singly tagged construct. However, we did not aim to optimize resolution in this work, so any further resolution optimization will be a topic for future work.

What changes need to be made to the acquisition parameters when using a triple versus a single tag construct?

We did not change the acquisition parameters between experiments using a triple versus a singly tagged construct, because our goal was to present a direct comparison between the two constructs. This would not be possible to do if we used different acquisition parameters for the two experiments.

8- "LIVE-PAINT enables longer data acquisition times". There are multiple concerns with this section.

A- The rationale for the design of this experiment isn't clear. Why use this approach (acquire, partial bleach, acquire) when you could simply acquire datasets for an extended period of time (~5 min) and then plotting the number of localizations versus time.

We used the acquire, partial bleach, acquire approach as opposed to extended imaging times in order to minimize the potential for exchange of the protein of interest that could occur with a long bleaching time.

Our aim was to adapt a FRAP approach to measure recovery in localization events in order to illustrate the ability of our reversible interaction pairs to exchange and better replenish signal compared with directly fusing a fluorescent protein to the protein of interest.

B- Show the datasets/images that were analyzed in the graph of Figure 3. What does the data look like?

The authors appreciate the feedback on the data formatting for Figure 3. We have supplemented the figure with maximum projection images for time ranges through the course of the experiment to provide evidence that the localizations obtained after bleaching continue to show specificity to the yeast septum. We have also reformatted

the raw data displayed in the figure to better show the reduced bleaching observed with our reversible interaction pairs compared to the direct fusion to a fluorescent protein. These changes are reflected in the updated Figure 3 in the paper.

The new Figure 3 legend text now reads: “Figure 3. LIVE-PAINT shows recovery of signal after bleaching. (A) LIVE-PAINT interaction pairs show more recovery in number of localization events than a direct fusion to a FP. In this experiment, fluorescence images were collected for 1,000 frames (50 s) at standard imaging power (3.1 W/cm^2), then the sample was photobleached using high laser power (26.6 W/cm^2), and then the sample was again imaged for 1,000 frames (50 s) at standard imaging power. Cdc12-SYNZIP18 + SYNZIP17-mNG (blue/green circles, each representing a single cell) retain many more localization events than Cdc12-mNG (gray circles, each representing a single cell) after two minutes of photobleaching. Each shade of gray or blue/green represents a single cell, which can be color-matched between pre-photobleaching (PrB) and post-photobleaching (PoB) conditions. DF = Cdc12-mNG (Direct Fusion); SZ = Cdc12-SYNZIP18 + SYNZIP17-mNG (SYNZIP pair). (B) Maximum projections for different frame ranges in both “before bleaching” and “after bleaching” videos demonstrate that signal obtained after bleaching continues to localize to the yeast septum. (Top) Maximum projections are shown for 200 frame ranges for a representative cell expressing Cdc12-mNG. (Bottom) Maximum projections are shown for a representative cell expressing Cdc12-SYNZIP18 + SYNZIP17-mNG. All “before bleaching” images are normalized to one another and, similarly, all “after bleaching” images are normalized to one another. Scale bar is $1 \mu\text{m}$.” (lines 293-310)

We have also added an additional supplementary figure (Figure S9, now referenced in lines 290-291) showing maximum projection images for ‘before’ and ‘after’ bleaching for

every cell analyzed in Figure 3. We hope this reveals to the reviewer the ability for our reversible interaction pairs to better replenish specific signal after bleaching.

C- If PAINT requires the fast binding and unbinding of the fluorescent label to the protein of interest, how can single emissions be captured for the directly labeled Cdc12 (data used for the black curve)? What do these single emissions look like?

We have added a sentence mentioning this point in the revised manuscript: “Although mNG is known to blink intrinsically¹⁹, we chose to use it in our experiments because it is very bright and therefore can produce very precise localization events” (lines 137-139). However, although there is likely a contribution from such blinking, the key effect that we are monitoring is due to ‘on/off’ binding of the peptide-binding-protein FP to the peptide that is fused to the protein of interest.

The key evidence supporting this statement is that our method works with different fluorescent proteins (mKO and mOrange). These fluorescent proteins are not known to blink intrinsically.

We have added an additional supplementary figure S1 (now referenced in lines 141-143) and two additional supplementary videos S1 and S2 which we have uploaded to Edinburgh DataShare with provided DOI: <https://doi.org/10.1101/2020.02.03.932228>.

5- The data in Figure S5A doesn't seem to show the stated effect. Are the data in each graph significantly different for the other? Superimposing the curves may show near perfect overlap.

We are grateful to the reviewer for motivating us to revisit the data in Figure 4 and S5 from the original manuscript. Prompted by these comments, we reanalysed the data and identified a bug in our data processing script. Specifically, we previously plotted one minus the cumulative probability distribution for localizations, as a function of time. This is not technically correct, because it implies that the localization rate goes to zero at the end of our video, even when it does not. Instead, we have now binned and summed the localizations, to get a number of localizations for each “time chunk” of a video. We then normalized these values to the highest value for each condition.

We have therefore corrected our data processing script, and we have now combined Figure 4 and (along with Figure S5 from the original manuscript) into a new Figure 4, to better present these data. Figure 4 shows that the data for different concentrations of galactose are clearly not the same. We see that as we increase the concentration of galactose (so that more peptide-binding-protein FP is expressed and consequently

there is a larger cytoplasmic reservoir for exchange) imaging can be continued for longer times. Quantitatively, we see that the exponential time constant (τ) varies from 1.9 s at 0% galactose to 139 s at 0.1% galactose, for the SYNZIP17-SYNZIP18 interaction pair. We also add an additional sentence to the text referencing Figure 4: “In Figure 4B, for example, we observe that when imaging Cdc12-SYNZIP18 + SYNZIP17-mNG using 0.1% galactose, even after 200 s of imaging, localizations are still being recorded at about 30-40% of the initial rate” (lines 326-329).

Figure 4. Localization rate decays more slowly with increased FP expression. Localization rate as a function of imaging time for (A) Cdc12-MEEVF + TRAP4-mNG and (B) Cdc12-SYNZIP18 + SYNZIP17-mNG, each at four different concentrations of galactose. Data for the MEEVF-TRAP4 interaction pair is for 0% galactose (red), 0.005% galactose (dark orange), 0.02% galactose (light orange) and 0.1% galactose (yellow). Data for the SYNZIP18-SYNZIP17 interaction pair is for 0% galactose (bright blue), 0.005% galactose (blue), 0.02% galactose (teal) and 0.1% galactose (mint). The data for each concentration of galactose were fit to a single exponential (shown as a solid line with matching color). For the MEEVF-TRAP4 interaction pair (A), the exponential time constant (τ) for the different concentrations of galactose is 0%: 4.7 s; 0.005%: 15 s; 0.02%: 32 s; 0.1%: 81 s. For the SYNZIP18-SYNZIP17 interaction pair (B), the exponential time constant (τ) for the different concentrations of galactose is: 0%: 1.9 s; 0.005%: 54 s; 0.02%: 73 s; 0.1%: 139 s. (lines 332-344).

7- Please provide a reasoning or reference for the following statement LINE 256 “This photobleaching reduces the resolution of the image because it limits the density of emitters that can be measured.” Cumulative density does not necessarily result in higher resolution, especially in live SMLM.

Higher localization density does result in higher image resolution (see Sharonov et al. PNAS, 2006).

10- Using LIVE-PAINT to label actin does not improve the currently used methods. LIVE-PAINT yields the exact same results as tagging actin directly with GFP. Actin tagged with GFP (or other tags as seen in Chen et al. J Struct Biol, 2012), is only incorporated into Arp2/3 polymerized actin patches and is excluded in all Formin polymerized actin filaments. Therefore LIVE-PAINT does not push the field forward in terms of tagging actin.

Our aim with this section was simply to demonstrate that using our method it is possible to label a protein that is typically difficult to label - ie actin. Our aim was to present a new tool, which others can use to ‘push the actin field forward’.

Two issues with the last full sentence in the legend of Figure 5 “Actin cables or rings are not observed either because we are imaging in TIRF or because the stringent structural requirements for actin in these structures means that even actin with very small ~2 kDa tags may be excluded from ring and cable structures²³”.

First, that statement should be in the text of the results section, not in a legend.

We have taken the reviewer’s advice and moved that statement into the text, and revised it: “Actin rings, or actin cables that span the cell, are likely not observed because we are imaging in TIRF, which illuminates only about 200 nm into the cell (a typical yeast cell is 1-3 μm thick).” (lines 362-364)

Second, the authors would have seen cables and rings if they had been present because they can focus through the entire thickness of a budding yeast cell even with TIRF.

This statement is not correct. TIRF does not focus through the entire thickness of a budding yeast cell. TIRF illuminates approximately 200 nm into the cell. A yeast cell is of the order 1-3 μm thick.

Consider deleting that part or provide an explanation why TIRF would prevent seeing cables and rings.

We have extended discussion of TIRF illumination volume with respect to actin structures and yeast cell size in the manuscript: “Alternatively, or additionally, it could be that the stringent structural requirements for actin in these assemblies means that even actin with very small ~2 kDa tags may be excluded from ring and cable structures²⁸” (lines 365-367).

11- LIVE-PAINT enables long tracking times in vivo. This data does not show the application of LIVE-PAINT to the measurements of protein dynamics using SMLM imaging technique in vivo. This work is done with diffraction limited microscopy, which is perfectly fine as it makes LIVE-PAINT a broadly applicable method, but it should be clearly stated in the text.

We thank the reviewer for identifying this lack of clarity in our manuscript. We have added some text to make clear that the tracking is performed with diffraction limited microscopy: “We therefore C-terminally tagged cofilin with SYNZIP18, and tracked it using the LIVE-PAINT strategy (diffraction-limited, not super-resolution)” (lines 390-392)

Why does the text not describe the data shown in the six different panels of Figure 6? This is beautiful analysis of particle diffusion but there’s no description or comparison with known kinetics of Cofilin.

As with our answer to point 11 above, regarding actin. Our aim with cofilin was to show that our method provides a new way to label cofilin, and to present it so that researchers who work on cofilin have another tool they can use.

The statement “We observed a wide range of behaviors” needs to be substantiated with data. What range of behaviors? Why are they relevant and important to this work?

We use the phrase, “We observed a wide range of behaviors”, because that is what we observed. We mean we observed a range of different diffusion rates, but these could well be associated with different association states of the protein. Our experiments were not intended to look into this point in depth, just to report what we saw.

Does labeling cofilin affects its function? Were controls performed to measure cell viability and actin network dynamics? This data needs to be shown.

There was no effect on either cell morphology or growth rate. This is not a paper about actin or cofilin. We just use these as example proteins. We hope that by presenting this method, those who work on actin and cofilin may be able to adopt it to increase the repertoire of methods available to them.

13- LINE 414. "...concurrent super-resolution imaging of multiple targets.". Showing the application of LIVE-PAINT with two spectrally distinct fluorescent proteins would increase the significance of this work for the field of SMLM. As the authors point out, SMLM imaging in live cells is restricted to a single color of fluorescence protein. New methods to expand the toolkit are needed. LIVE-PAINT could offer an option but without proof of applicability, option of co-expressing two fluorescent constructs at the appropriate cellular concentrations, this remains only a hopeful thought.

Dual color imaging is beyond the scope of this paper. This paper is the first report of our method, and we seek only to show it works, to show some data on how performance changes as we change parameters, and to disseminate it to other researchers so they can adopt it in their own research.

To address this point, however, we have added to the supplementary information data on different peptide-protein pairs (Figure S2, now referenced in lines 150-154) and also data on using the method with different fluorescent proteins (Figure S1, now referenced in lines 141-143). Additionally, we have supplied data showing co-expression and imaging of two protein targets using two orthogonal interaction pairs and two spectrally distinct fluorescent proteins, mNeonGreen and mCherry (Figure S4, now referenced in lines 150-154). We believe that this additional data substantiates our point, and shows that all the components are in place for multi-colour super-resolution imaging in the future.

14- LINE 452. "Two of the three peptide-pairs that we tested were suitable for LIVE-PAINT." Three techniques weren't shown. Wouldn't it be useful to describe the three tested options and explain why one didn't work?

We have expanded this section in the text to more fully describe the three tested pairs, and the problems we encountered with the one pair, which explain why we did not continue using it. Briefly, the third interaction pair we did not continue using in our experiments was a charged coiled-coil pair CC-B_N^{3.5}-CC-A_N^{3.5} (Thomas et al. JACS, 2013), with the CC-B_N^{3.5} half fused to the protein of interest and the CC-A_N^{3.5} half fused to a fluorescent protein. This interaction pair has approximately 5 nM affinity (as measured *in vitro*). The vast majority of cells expressing CC-B_N^{3.5} fused to CDC12 exhibited an irregular morphology. The cells were elongated and appeared to have

difficulty dividing properly. We have added a new methods section to more clearly describe our process for selecting the interaction pairs we used in our work.

Our rationale reads: “TRAP4-MEEVF¹⁵, SYNZIP17-SYNZIP18¹⁷, and CC-A_N^{3.5}-CC-B_N^{3.5}³³ interaction pairs were identified in the literature and tested *in vivo* in live yeast by fusing one half of the interaction pair to a protein of interest and the other half to a FP. The cells were imaged under a microscope as described in the “Microscopy” section. The TRAP4-MEEVF and SYNZIP17-SYNZIP18 interaction pairs showed no morphological abnormalities, no noticeable growth defect in liquid culture or on plates and showed fluorescence at the expected locus of the tagged protein. For this reason, these two interaction pairs were used in this work. Using CC-A_N^{3.5}-CC-B_N^{3.5} interaction pair resulted in unusual cell morphology in the vast majority of cells, with elongated cell shapes, so this pair was not used further. The 101A-101B and 108A-108B interaction pairs¹⁸ were also checked and did not cause any changes to cell morphology or changes to cell growth.” (lines 512-522).

15- *Why were cells imaged in water? Is there no effect of keeping budding yeast cells in water rather than in medium? One would think that the change in the osmolarity would affect key molecular processes such as endocytosis.*

We grew the cells in minimal media and washed the cells into water immediately before imaging. Yeast is very tolerant to osmolarity changes. It is even standard in yeast transformation procedures to do multiple washes in water and cells are still viable after this procedure. For these reasons, we determined it was suitable to image the cells in water, which is useful because it has very little autofluorescence.

Minor issues

3- *Figure 1C. Add a line showing threshold photon value on the graph to identify the noise from the positive localizations.*

The algorithm we use in ImageJ to distinguish background fluorescence from true localizations is more complicated than simply using a photon threshold, so we cannot add this line to our figure. See the methods section for more information about the ImageJ plugins we used to identify localization events.

16- *Figure 2. Why is there cell wall/cell periphery labeling in the TRAP 0% image?*

This cell appears to simply have a large vacuole, which effectively pushes the cytoplasm (and any fluorescent molecules it contains) towards the outside of the cell.

On close inspection, it is evident that the 'labeling' is not really around the edge of the cell, and does not quite look like what you would expect to see in cells where the cell wall or plasma membrane is labeled.

Reviewers' comments:

Reviewer #1 (Remarks to the Author):

The Manuscript by Oi et al. has been revised thoroughly. The Referee is satisfied by the rebuttal and the additional data/clarifications included with the revision. Some discrepancies were eliminated by re-analysing the data (Figure 4). At this point, the Referee would refrain from any further critique of the Manuscript, since the quality of scientific presentation and the data merit the publication. The rest should be left to the judgement of the reader.

The Referee suggests the followed minor amendments:

[1] Please add some visual clues to improve the visual presentation of Figure 4. An arrow indicating the direction of change in the concentration of the galactose would be sufficient (and would also help colour-blinded readers)

[2] Please add imaging details (laser power, exposure) for nanoscopy-related figures (Figures S1, S2)

Reviewer #2 (Remarks to the Author):

The authors answered raised questions and adopted some changes. The manuscript was further improved by the authors, but the authors did not provide enough information how several quantifications were performed. Therefore, data quality and reproducibility remains hard to assess. 1.) The authors should give sufficient n numbers and statistics when an important conclusion is made. I agree with the authors that this is not absolutely necessary in Figure S5 (expression) since an established system was used. Figure 3A (recovery of signal after bleaching) is transparent because individual values of a larger number of experiments is presented. The conclusion from Figure 4A and B (Localization rate decays more slowly with increased FP expression) might be plausible because the concentration dependency is visible in the whole set of experiments (but already borderline). Figure S7 makes the conclusion that 3xmNG has an improved localization precision to 1xmNG. This conclusion should be supported by a statistically significant difference in localization precision. The presentation with the same box or triangle and color is not clear (which box is belonging to which measurement). Especially the 3x mNG measurement looks quite variable making a larger n and statistics more eligible.

Also Figure S8 remains untransparent. How many localizations were analyzed? Independent experiments? Are the results reproducible or just due to an unhealthy cell (for example with changed expression level) or an accidentally changed microscope setting in the one performed experiment? Figure S10 is also intransparent. Nice to have a boxplot (no bar diagram) but quantification on just one image quantified for each video length? The figure legend is not conclusive. I think there is also a type error in the first sentence (off at?).

Other researchers will adapt this method to their requirements and need a validated strategy to follow. The authors themselves declare that introduction of a new tool is the major aim of the study.

2.) Supplementary data were hard to find, especially the movies (with huge data files). Attaching a direct link for the reviewers would be more intuitive <https://datashare.is.ed.ac.uk/handle/10283/3610>. For readers a deposition at the journal homepage for example as mp4 is important.

3.) In Figure 4 "0.05" is written below the figure. Is that a remainder of an older figure version?

I think there are still minor revisions necessary in data presentation and transparency and possibly repetitions of some experiments (which is currently hard to assess due to lacking information). The journal editor should have and give a clear guideline for n numbers, statistics, and transparency to ensure reproducibility. The study remains interesting but it is not sure if every reader will consider the data as reliable in the current version.

Reviewer #3 (Remarks to the Author):

This work describes LIVE-PAINT, a new method for labeling proteins in live cells, derived from the principles of DNA-PAINT and using the PAINT imaging method. LIVE-PAINT has the potential to expand our toolbox of available fluorescent labels for SMLM that is currently very restrictive for live imaging. The LIVE-PAINT method described by Oi et al. is functional and applicable.

Major concerns raised about this manuscript remain after the first round of reviews. These concerns were not addressed experimentally or by modifying the text. The same critical concerns are listed below and one additional concern is added. Because they are major concerns, they raise flaws with the manuscript that can lead to misinterpretations by readers. Many of the other concerns were addressed.

Specific comments, major conceptual concerns:

1- The same major flaw remains after the first revision that there is no application of LIVE-PAINT that support the title of Result section 1 "LIVE-PAINT achieves super-resolution inside live cells using reversible peptide-protein interactions".

2- The lack of optimization of the acquisition and analysis parameters are major concerns about this work. The authors mention that running the camera faster would not be advantageous to their work.

Excerpt from rebuttal:

Importantly, can you run the camera at a faster frame rate? EMCCD cameras can reach faster frame rates if the field of view is restricted to a smaller frame.

- Although it is possible to run at a faster frame rate, there is no advantage to doing so. The localisation precision is dependent on the number of photons collected. The time it takes to collect the optimum number of photons is dependent on the photobleaching lifetime of the fluorescent protein, which should match, as closely as possible, the time it remains bound to its target protein. We use a low excitation power to reduce photodamage, it is this time that limits how fast we can image, and not the frame rate of the camera.

This implies that resolution will be lost to the blurring due to the dynamics that occur in live cells. Such effect of dynamics needs to be addressed in the text.

3- The section about tagging actin is confusing. In the title, you mention "direct fusion", which implies that you'll be writing about the direct tagging of actin with a fluorescent protein. But then you refer to Courtemanche et al., which is about indirect tagging of actin using Lifeact. If you want to describe all work on actin (both direct and indirect labeling), you need to include references from the Pollard lab (Arasada et al, 2018 and Laplante et al, 2016) where live FPALM imaging using mEos3.2-CHD was performed, especially Arasada et al. MBoC, 2018 as it refers to actin patches. Also, you must remove

the following sentence "Moreover, very few of these methods can be used inside live cells and none is currently compatible with live cell super-resolution imaging.". Please rephrase the section to acknowledge that indirect labeling has been successfully applied to live superresolution imaging and your approach offers a direct labeling alternative.

Our point by point response to the reviewer comments.
Reviewer comments are in plain text, our responses are in *italics*.

Reviewer #1 (Remarks to the Author):

The Manuscript by Oi et al. has been revised thoroughly. The Referee is satisfied by the rebuttal and the additional data/clarifications included with the revision. Some discrepancies were eliminated by re-analysing the data (Figure 4). At this point, the Referee would refrain from any further critique of the Manuscript, since the quality of scientific presentation and the data merit the publication. The rest should be left to the judgement of the reader.

We thank the reviewer for their positive response to our changes, and their favourable opinion of the modified manuscript. We appreciate their conclusion that the manuscript merits publication.

The Referee suggests the followed minor amendments:

[1] Please add some visual clues to improve the visual presentation of Figure 4. An arrow indicating the direction of change in the concentration of the galactose would be sufficient (and would also help colour-blinded readers)

We thank the reviewer for this suggestion. We have modified the figure to include a legend indicating the colour associated with each galactose concentration. The different shades, in the same colour family, can be distinguished by colour-blind readers. We opted against adding an arrow, because it would need to cross the other lines in the figure and would add clutter.

[2] Please add imaging details (laser power, exposure) for nanoscopy-related figures (Figures S1, S2)

These details have now been included in all relevant SI figure legends.

Reviewer #2 (Remarks to the Author):

The authors answered raised questions and adopted some changes. The manuscript was further improved by the authors, but the authors did not provide enough information how several quantifications were performed. Therefore, data quality and reproducibility remains hard to assess.

1.) The authors should give sufficient n numbers and statistics when an important conclusion is made. I agree with the authors that this is not absolutely necessary in Figure S5 (expression) since an established system was used.

OK

Figure 3A (recovery of signal after bleaching) is transparent because individual values of a larger number of experiments is presented.

The conclusion from Figure 4A and B (Localization rate decays more slowly with increased FP expression) might be plausible because the concentration dependency is visible in the whole set of experiments (but already borderline).

OK

Figure S7 makes the conclusion that 3xmNG has an improved localization precision to 1xmNG. This conclusion should be supported by a statistically significant difference in localization precision. The presentation with the same box or triangle and color is not clear (which box is belonging to which measurement). Especially the 3x mNG measurement looks quite variable making a larger n and statistics more eligible.

We compared the mean and median precision values for the two 1xmNG and 3xmNG replicates via a t-test. The mean precision values for the 1xmNG replicates was 63.4 nm and 63.9 nm, while the mean precision values for the 3xmNG replicates was 54.5 nm and 55.3 nm. A t-test indicated that these two sets of measurements was statistically significant, with a p-value of 0.05. When comparing the medians (1xmNG: 63.1 nm and 63.5 nm; 3xmNG: 52.2 nm and 53.1 nm), we obtained a similar result, with a p-value of 0.038.

We include these data to illustrate to the reader the potential of this approach. We tested just one version of an mNG tandem construct. To increase the effect of using tandem fluorescent proteins, optimization work that is beyond the scope of the paper would be needed. The ideal tandem array would be expressed as well as a single mNG, all the individual proteins in a tandem array would fold correctly, and the tandem fluorescent proteins would be a suitable distance apart to avoid any mutual quenching. Because the tandem arrays are not the main focus of the paper, none of these possible optimizations are included. Rather, our goal is to alert

the reader to the possibility of using tandem arrays to improve imaging of whatever system they chose.

We have also updated the supplementary figure to use different shapes (circles and squares) for the two replicates and to draw line plots through each set of points for the replicates. We note, however, that even though the replicates look somewhat variable the mean and median precision values of the replicates are not particularly variable. The quantitative difference between the 1xmNG and 3xmNG replicates is perhaps more easily visualized by a cumulative distribution of the precision values as shown below.

Also Figure S8 remains untransparent. How many localizations were analyzed? Independent experiments? Are the results reproducible or just due to an unhealthy cell (for example with changed expression level) or an accidentally changed microscope setting in the one performed experiment?

We have added the details that the reviewer requests to the legend of Figure S8 (which is now Figure S9). The data represent two technical replicates for each of the 1xmNG and 3xmNG strains. Since multiple cells (two to six) were present in each field of view, the results are not due to any anomaly with a single outlier cell. Fields of view were chosen by visual inspection to not include dead cells (dead cells were very uncommon, we estimate less than 1%). Also, since the experiments were performed on the same day along with other experiments, the likelihood of an accidentally changed microscope setting giving rise to the difference is negligible, because other experiments performed on the same day do not show the same difference.

Figure S10 is also intransparent. Nice to have a boxplot (no bar diagram) but quantification on just one image quantified for each video length? The figure legend is not conclusive. I think there is also a type error in the first sentence (off at?).

Other researchers will adapt this method to their requirements and need a validated strategy to follow. The authors themselves declare that introduction of a new tool is the major aim of the study.

We agree with the reviewer that the figure is not particularly clear and upon reflection we do not think it provides critical support to the findings in the paper. We have therefore decided to remove Figure S10 from the manuscript. We have replaced it with a similar but improved supplementary figure (now Figure S2). This figure looks at the resolution of our Cdc12 images for different data collection times. It compares the number of localizations in the septum to the resolution of the image and shows that we can obtain approximately 20 nm resolution within about 5 s of imaging. We have adjusted the numbering for all the supplementary figures accordingly. We have also removed the sentence referencing Figure S10 in the manuscript. We thank the reviewer for helping us critically evaluate the content of this figure. The updated figure is shown below.

2.) Supplementary data were hard to find, especially the movies (with huge data files). Attaching a direct link for the reviewers would be more intuitive

<https://datashare.is.ed.ac.uk/handle/10283/3610>. For readers a deposition at the journal homepage for example as mp4 is important.

We apologise for the difficulty in accessing these files. In addition to depositing data at datashare, we have also generated compressed .mp4 files and will deposit with the journal. All the data will also be available at datashare, as before, with the DOI specified in the paper - <https://doi.org/10.1101/2020.02.03.932228>.

3.) In Figure 4 “0.05” is written below the figure. Is that a remainder of an older figure version?

We thank the reviewer for drawing our attention to this residue, which has now been removed.

I think there are still minor revisions necessary in data presentation and transparency and possibly repetitions of some experiments (which is currently hard to assess due to lacking information). The journal editor should have and give a clear guideline for n numbers, statistics, and transparency to ensure reproducibility. The study remains interesting but it is not sure if every reader will consider the data as reliable in the current version.

We have included n numbers in each figure legend, for the data shown, and have also performed statistical tests where necessary.

Reviewer #3 (Remarks to the Author):

This work describes LIVE-PAINT, a new method for labeling proteins in live cells, derived from the principles of DNA-PAINT and using the PAINT imaging method. LIVE-PAINT has the potential to expand our toolbox of available fluorescent labels for SMLM that is currently very restrictive for live imaging. The LIVE-PAINT method described by Oi et al. is functional and applicable.

We thank the reviewer for their positive remarks about the functionality and applicability of our method.

Major concerns raised about this manuscript remain after the first round of reviews. These concerns were not addressed experimentally or by modifying the text.

We are surprised at this comment because we made major revisions to both the text and figures based on the first round of reviewer comments. Both reviewer 1 and 2 recognize we have made substantial revisions and that the manuscript is now suitable for publication.

Reviewer 1: “The Manuscript by Oi et al. has been revised thoroughly. The Referee is satisfied by the rebuttal and the additional data/clarifications included with the revision.”

Reviewer 2: “The authors answered raised questions and adopted some changes.”

The same critical concerns are listed below and one additional concern is added. Because they are major concerns, they raise flaws with the manuscript that can lead to misinterpretations by readers. Many of the other concerns were addressed.

Specific comments, major conceptual concerns:

1- The same major flaw remains after the first revision that there is no application of LIVE-PAINT that support the title of Result section 1 “LIVE-PAINT achieves super-resolution inside live cells using reversible peptide-protein interactions”.

The title exactly describes what the paper is about. We have demonstrated super-resolution imaging within live cells using peptide-protein interactions, typically achieving a resolution of ~20 nm.

2- The lack of optimization of the acquisition and analysis parameters are major concerns about this work. The authors mention that running the camera faster would not be advantageous to their work.

Excerpt from rebuttal:

Importantly, can you run the camera at a faster frame rate? EMCCD cameras can reach faster frame rates if the field of view is restricted to a smaller frame.

- Although it is possible to run at a faster frame rate, there is no advantage to doing so. The localisation precision is dependent on the number of photons collected. The time it takes to collect the optimum number of photons is dependent on the photobleaching lifetime of the fluorescent protein, which should match, as closely as possible, the time it remains bound to its target protein. We use a low excitation power to reduce photodamage, it is this time that limits how fast we can image, and not the frame rate of the camera.

This implies that resolution will be lost to the blurring due to the dynamics that occur in live cells. Such effect of dynamics needs to be addressed in the text.

Resolution lost due to the dynamics of a protein of interest is not a feature specific to our method. It is a general limitation of all super-resolution imaging methods. It would not be appropriate to provide a general review of the limitations and strengths of all super-resolution in this manuscript.

While we appreciate that choosing a slow frame rate would result in blurring, choosing a frame rate that is too fast would result in insufficient photons collected to obtain high precision localization events. The frame rate we chose for our experiments represents a happy medium between these two extremes. More than this, optimization of frame rate is beyond the scope of this work. The frame rate would need to be individually optimized for each protein studied and each fluorescent protein used for imaging.

Additionally, we focus our imaging on proteins which localize to discrete structures inside the cell, limiting this resolution loss in our case. While we appreciate that choosing a slow frame rate would result in blurring, choosing a frame rate that is too fast would result in insufficient photons collected to obtain high precision localization events. The frame rate we chose for our experiments fits this happy medium. Rigorous optimization of frame rate falls outside the scope of this work, as it would need to be optimized for each individual protein to be studied and each fluorescent protein used for imaging.

3- The section about tagging actin is confusing. In the title, you mention “direct fusion”, which implies that you’ll be writing about the direct tagging of actin with a fluorescent protein.

But then you refer to Courtemanche et al., which is about indirect tagging of actin using Lifeact. If you want to describe all work on actin (both direct and indirect labeling), you need to include references from the Pollard lab (Arasada et al, 2018 and Laplante et al, 2016) where live FPALM imaging using mEos3.2-CHD was performed, especially Arasada et al. MBoC, 2018 as it refers to actin patches.

We thank the reviewer for recommending these references. We have added them to this section of the paper. (lines 357-358)

Also, you must remove the following sentence “Moreover, very few of these methods can be used inside live cells and none is currently compatible with live cell super-resolution imaging.”. Please rephrase the section to acknowledge that indirect labeling has been successfully applied to live superresolution imaging and your approach offers a direct labeling alternative.

We have rephrased this section as follows:

“Direct fusion of actin to the photoconvertible fluorescent protein mEos, expressed alongside unmodified actin, has been used to image actin using PALM (Arasada et al, 2018 and Laplante et al, 2016). The mEos protein is a rather large addition to actin, and undoubtedly results in some perturbation of function (as evidenced by cells expressing only actin-mEos, in the absence of any unmodified actin, being unviable). LifeAct is a peptide that binds to the polymerized form of actin, and not the unpolymerized form. The perturbation to the equilibrium distribution of actin forms that LifeAct causes has been noted (Courtemanche et al. 2016). Nevertheless, the binding and unbinding of LifeAct has been used to image actin filaments in live cells (Kiuchi et al. 2015) (using a PAINT-like methodology). We note and reference this result, however polymerized actin is the only protein that can be imaged using LifeAct, our method can be applied to any protein - including actin - and we present its application to actin to provide another possible tool for actin researchers.” (lines 357-368)